# Mutant Lef1 controls Gata6 in sebaceous gland development and cancer

Bénédicte Oulès[1] , Emanuel Rognoni[1,2] , Esther Hoste[1,3,4] , Georgina Goss[1], Ryan Fiehler[5], Ken Natsuga[6], Sven Quist[7], Remco Mentink[8], Giacomo Donati[1,9,†] & Fiona M Watt[1,*,†]

## Abstract

Mutations in Lef1 occur in human and mouse sebaceous gland (SG) tumors, but their contribution to carcinogenesis remains unclear. Since Gata6 controls lineage identity in SG, we investigated the link between these two transcription factors. Here, we show that Gata6 is a β-catenin-independent transcriptional target of mutant Lef1. During epidermal development, Gata6 is expressed in a subset of Sox9-positive Lef1-negative hair follicle progenitors that give rise to the upper SG. Overexpression of Gata6 by *in utero* lentiviral injection is sufficient to induce ectopic sebaceous gland elements. In mice overexpressing mutant Lef1, Gata6 ablation increases the total number of skin tumors yet decreases the proportion of SG tumors. The increased tumor burden correlates with impaired DNA mismatch repair and decreased expression of Mlh1 and Msh2 genes, defects frequently observed in human sebaceous neoplasia. Gata6 specifically marks human SG tumors and also defines tumors with elements of sebaceous differentiation, including a subset of basal cell carcinomas. Our findings reveal that Gata6 controls sebaceous gland development and cancer.

Keywords DNA mismatch repair; Gata6; Lef1; sebaceous gland; sebaceous tumors
Subject Categories Cancer; Development & Differentiation; Transcription
The EMBO Journal (2019) 38: e100526

## Introduction

At the core of the canonical Wnt signaling pathway is nuclear translocation of β-catenin and consequent transcription of downstream target genes through β-catenin binding to members of the lymphoid enhancer-binding factor/T-cell factor (Lef/Tcf) transcription factor family (Klaus & Birchmeier, 2008; Nusse & Clevers, 2017). The Wnt pathway plays a central role in stem cell maintenance and fate specification in mammalian epidermis (Watt & Collins, 2008; Lim & Nusse, 2012), controlling the balance between hair follicle (HF) and sebaceous gland (SG) differentiation. During embryonic life and during the postnatal hair cycles, activation of β-catenin triggers HF growth (Huelsken *et al*, 2001; Lowry *et al*, 2005; Donati *et al*, 2014). Ectopic HF can also be generated upon transient activation of epidermal β-catenin (Lo Celso *et al*, 2004; Silva-Vargas *et al*, 2005), in particular in Lrig1-positive and Lgr6-positive stem cell (SC) populations of the upper pilosebaceous unit (Kretzschmar *et al*, 2016). While Wnt signaling favors HF over SG fate, an N-terminally truncated form of Lef1 (ΔNLef1), unable to bind β-catenin, converts HF into keratinized epidermal cysts with ectopic sebocytes (Merrill *et al*, 2001; Niemann *et al*, 2002; Donati *et al*, 2017).

Deregulation of the Wnt/β-catenin pathway occurs in several skin cancers. Transgenic mice overexpressing a stabilized form of β-catenin develop HF tumors: pilomatricomas or trichofolliculomas (Gat *et al*, 1998; Lo Celso *et al*, 2004). In humans, stabilizing mutations in β-catenin are found in a majority of pilomatricomas (Chan *et al*, 1999) and pilomatrix carcinomas (Lazar *et al*, 2005).

While genetic deletion of β-catenin from the epidermis is not associated with tumor development (Huelsken *et al*, 2001; Malanchi *et al*, 2008), transgenic mice expressing ΔNLef1 under the control of the keratin 14 promoter (K14ΔNLef1) spontaneously develop skin tumors, most of which are sebaceous adenomas and sebaceomas (Niemann *et al*, 2002). In K14ΔNLef1 mice, ΔNLef1 decreases endogenous Lef1 expression and acts as a dominant negative inhibitor of β-catenin (Niemann *et al*, 2002). Therefore, Wnt/β-catenin targets are downregulated (Donati *et al*, 2017) and the hair follicle cycle is compromised (Niemann *et al*, 2002). The ΔNLef1 transgene acts as a tumor promoter in chemical carcinogenesis experiments. Tumors that develop upon ΔNLef1 expression exhibit sebaceous

1 Centre for Stem Cells and Regenerative Medicine, King's College London, London, UK
2 Centre for Endocrinology, William Harvey Research Institute, Barts and the London School of Medicine and Dentistry, Queen Mary University of London, London, UK
3 Unit for Cellular and Molecular Pathophysiology, VIB Center for Inflammation Research, Ghent, Belgium
4 Department of Biomedical Molecular Biology, Ghent University, Ghent, Belgium
5 F-star Biotechnology Limited, Cambridge, UK
6 Department of Dermatology, Hokkaido University Graduate School of Medicine, Sapporo, Japan
7 Clinic for Dermatology and Venereology, Otto-von-Guericke-University, Magdeburg, Germany
8 Bejo, Warmenhuizen, The Netherlands
9 Department of Life Sciences and Systems Biology, Molecular Biotechnology Center, University of Turin, Turin, Italy
*Corresponding author. Tel: +44 2071885608; E-mail: fiona.watt@kcl.ac.uk
†These authors contributed equally to this work

 

differentiation rather than the papillomas and squamous cell carcinomas characteristic of wild-type (WT) mice (Niemann *et al*, 2007). Consistent with these findings, mutations in the N-terminus of Lef1 that prevent β-catenin binding are found in approximately 30% of human benign sebaceous tumors (Takeda *et al*, 2006) and 20% of eyelid sebaceous carcinomas (Jayaraj *et al*, 2015).

We recently reported that the transcription factor Gata6 plays a role in sebaceous lineage determination and is highly upregulated in the junctional zone (JZ) of K14ΔNLef1 mice (Donati *et al*, 2017). Therefore, in the present work, we sought to explore the role of Gata6 in establishing the sebaceous lineage and to understand the link between Gata6 and Lef1 in the context of sebaceous gland tumors. Our findings demonstrate that sebaceous fate during cancer is controlled by Lef1 and Gata6.

## Results

### Gata6 is a ΔNLef1 target gene

As reported previously (Merrill *et al*, 2001; Niemann *et al*, 2002; Petersson *et al*, 2011, 2015), in adult K14ΔNLef1 mice, the HF undergoes conversion to multilayered, keratinized cysts with ectopic sebocytes (Fig 1A and B). Based on our earlier studies, we now believe that these cysts represent abnormal Gata6-positive sebaceous ducts (SD) while the associated mature sebocytes do not express Gata6 (Donati *et al*, 2017). To identify direct ΔNLef1 target genes, we performed chromatin immunoprecipitation and next-generation sequencing (ChIP-Seq) on keratinocytes isolated from K14ΔNLef1 epidermis (Fig 1C). As expected, the intact ΔNLef1 DNA binding domain bound the same DNA core motif as Lef1, as, for example, on Vgll4 and Runx1 loci (Fig 1C).

To understand the transcriptional role of ΔNLef1 in the epidermis, we compared ΔNLef1 direct target genes to the gene signatures of the three major epidermal compartments (interfollicular epidermis (IFE), HF, and SG) obtained upon micro-dissection of adult tail skin from WT mice (Fig 1D and E; Donati *et al*, 2017). We observed

that ΔNLef1 mainly bound to repressed genes belonging to the SG signature (Fig 1D and E), consistent with a known TLE/Groucho-dependent repressive function of Lef1 in the absence of β-catenin (Niemann *et al*, 2002; Ramakrishnan *et al*, 2018).

Further analysis of the ChIP-Seq data revealed that Gata6 was a direct transcriptional target of ΔNLef1 (Fig 1C). We confirmed by RT–qPCR that Gata6 expression was increased in K14ΔNLef1 keratinocytes as compared to WT cells (Fig 1F). RNAi-mediated knockdown of Lef1 led to a striking Gata6 downregulation (Fig 1F), while ΔNLef1 overexpression in the SebE6E7 sebocyte cell line increased Gata6 expression (Fig 1G). These results are in agreement with the expansion of Gata6-positive cells observed in K14ΔNLef1 mice (Fig 1H; Donati *et al*, 2017).

To uncover the *in vivo* changes in gene expression linked to ΔNLef1 expression, we compared the gene expression profiles of flow-sorted total basal keratinocytes (basal, Itga6$^+$Cd34$^-$) and bulge stem cells (HFSC, Itga6$^+$Cd34$^+$) in WT and K14ΔNLef1 transgenic mice (Fig EV1A). To detect early molecular events, we collected cells from 9.5-week-old mice, when the HFs are in the resting (telogen) phase of the hair growth cycle (Oh *et al*, 2016), and the ΔNLef1 phenotype is not yet fully apparent (Niemann *et al*, 2002). Differentially expressed genes (DEG) between the different keratinocyte populations were validated by RT–qPCR (Fig EV1B). ΔNLef1 expression favored gene expression characteristic of the pilosebaceous unit while repressing the IFE gene signature (Fig 1I), and differently impacted several signaling pathways in the three compartments (Fig EV1C). The correlation between bulge and SG signatures in K14ΔNLef1 mice significantly increased when we selected genes that were direct targets of ΔNLef1 (white box in Fig 1J), suggesting a direct role of ΔNLef1 in the formation of ectopic SG in the HF. In addition, Gene Ontology (GO) of DEG in WT as compared to K14ΔNLef1 cells showed enrichment for the terms "tissue morphogenesis" and "gland development" (Fig 1K).

We next intersected ΔNLef1 direct target genes with DEG in SG (vs. IFE and HF) and with K14ΔNLef1 (vs. WT) epidermis transcriptomes (Fig EV1D). In addition to Gata6 (Fig 1C), ΔNLef1 directly upregulates Pparg and Edar, two well-characterized positive

---

**Figure 1.  ΔNLef1 drives Gata6 expression and SG transcriptional signature.**

A   Hematoxylin and eosin (H&E) staining of WT and K14ΔNLef1 HF from back skin at 11.5 weeks (early anagen). White arrowheads: ectopic SG and epidermal cysts.

B   Tail epidermal whole mounts from WT and K14ΔNLef1 mice labeled with anti-Fabp5, Krt15, and counterstained with Dapi.

C   Average signal intensity of ΔNLef1 binding sites detected by a Lef1 antibody (red) as compared to input alone (gray) and negative control (Neg. Ctl., black). DNA motif analysis revealed Lef1 classical consensus sequence. Representative plots of ChIP-Seq reads aligned to the Vgll4, Runx1, and Gata6 loci.

D   Heat-map (Pearson's correlation) of differentially expressed genes (DEG) between different micro-dissected regions (IFE, HF, and SG) from WT mice (left panel) (Donati *et al*, 2017). ΔNLef1 Chip-Seq data show ΔNLef1 direct transcriptional targets in black (right panel).

E   Gene Set Enrichment Analysis comparing gene location of ΔNLef1 peaks and SG gene expression signature.

F   RT–qPCR analysis of Gata6 in WT and K14ΔNLef1 primary keratinocytes transfected with siRNA targeting ΔNLef1 or a scrambled sequence (scr.). Data are means ± SEM of three independent wells.

G   RT–qPCR analysis of Gata6 in human SebE6E7 sebocytes transfected for 48 h with a mock plasmid or a Lef1- or ΔN34Lef1-expressing plasmid. ΔN34Lef1 is the human ortholog of murine ΔN32Lef1, which is expressed in K14ΔNLef1 transgenic mice (Takeda *et al*, 2006). Data are means ± SEM of three independent experiments.

H   Dorsal skin sections of WT and K14ΔNLef1 mice stained for Krt14 and Gata6.

I   Heat-maps of DEG from WT- or K14ΔNLef1-unfractionated (all) or sorted (Itga6$^+$Cd34$^-$, basal; Itga6$^+$Cd34$^+$, HFSC) keratinocytes in comparison with IFE, SG, and HF gene signatures ranked from high to low expression.

J   Heat-map depicting similarities, as Pearson's correlation coefficient, between DEG in keratinocytes from WT or K14ΔNLef1 mice and IFE vs SG vs HF transcriptome analysis (left panel). Additional correlation with ΔNLef1 Chip-Seq data is depicted in the right hand heat-map.

K   GO enrichment analysis of ΔNLef1 direct target genes among DEG of WT and K14ΔNLef1 mice.

Data information: (A, B, and H) Scale bars: 50 μm. (F, G) Statistical analyses were performed with an ordinary one-way ANOVA: (ns) not significant; *$P < 0.05$; ***$P < 0.0005$.

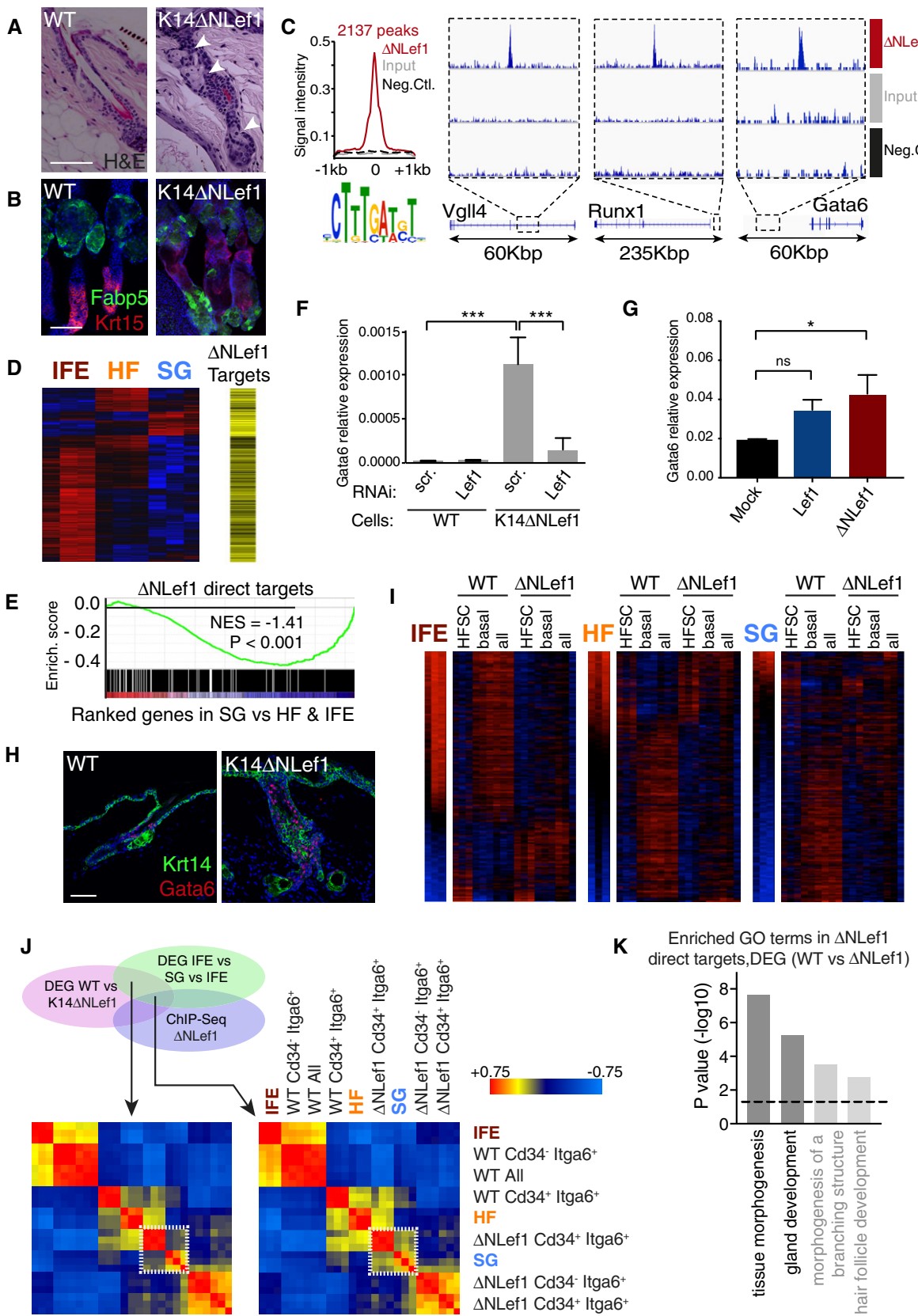

**Figure 1.**

regulators of the SG (Chang *et al*, 2009; Niemann & Horsley, 2012). In parallel, ΔNLef1 directly downregulates genes such as Klf5, important for IFE identity (Ge *et al*, 2017), Igfbp3 that is expressed in SG (Dahlhoff *et al*, 2016), and Slco2a1. Loss-of-function mutations in Slco2a1 are associated with sebaceous hyperplasia (Guo *et al*, 2017). We conclude that ΔNLef1 has activating and repressive functions that converge in promoting a SD/SG phenotype in K14ΔNLef1 mice.

### Gata6 expression is not dependent on Lef1 or β-catenin during epidermal development

Although Gata6 is a ΔNLef1 transcriptional target, ΔNLef1 did not trigger ectopic Gata6 expression in the IFE of K14ΔNLef1 mice (Fig 1H), suggesting that the upper pilosebaceous compartment is the only permissive niche for ΔNLef1-mediated Gata6 expression. During early HF/SG morphogenesis (from E15.5 to E18.5), Gata6 and Lef1 were not co-expressed in epidermal cells (Fig 2A), and at E18.5, Lef1$^{-/-}$ epidermal Gata6 expression was indistinguishable from control epidermis (Fig 2B). In embryonic WT skin, endogenous Gata6 appeared mainly in suprabasal cells located in the upper part of stage 4 HF (late hair peg; Fig 2A). While Gata6 and Lef1 did not co-localize in developing HF, Gata6 expression was detected in a subset of Sox9-positive cells in stage 4 and 5 HF (Fig 2A). In anagen HF of adult WT mice, endogenous Lef1 and Gata6 did not co-localize (Fig EV2A).

From birth, Gata6 was expressed, as in the adult epidermis, in the JZ and upper SG, but not in mature sebocytes (Fig EV2B). Consistent with the restriction of Sox9 expression to bulge cells in adult HF (Nowak *et al*, 2008), there was little co-expression of Gata6 and Sox9 at P1 (Fig EV2B). Gata6-positive cells did not co-localize with another bulge marker, CD34. Cells expressing low levels of Gata6 co-expressed Lgr6 and Lrig1 in neonatal P1 mice, as in adult animals (Donati *et al*, 2017). No co-expression of Gata6 and Tcf3/4 was observed. Gata6 did not co-localize with Ki67 (Fig EV2B), indicating that Gata6-positive cells were not proliferative, in agreement with the role of Gata6 in terminal differentiation of the SD lineage (Donati *et al*, 2017). During HF morphogenesis in human embryonic skin, Gata6 expression was initiated at a similar HF stage to the mouse and was located in the JZ, SD, and upper SG of more mature HF (Fig EV2C).

To test whether Gata6 expression was regulated by canonical Wnt signaling, we used inducible and constitutive epidermal β-catenin gain-of-function mouse models (Fig 2C and D). K14ΔNβ-CateninER mice in which N-terminally truncated β-catenin is fused with the estrogen receptor ligand binding domain (ER) (line D2, 12

copies; line D4, 21 copies) served as the inducible model (Lo Celso *et al*, 2004). Upon application of tamoxifen (4OHT) for 1 or 6 days, HF entered anagen in both mouse lines. In the D4 line, this was accompanied by massive thickening of the existing HF and induction of ectopic follicles. However, we did not observe any ectopic expression of Gata6 (Fig 2C).

For constitutive β-catenin activation, we generated K14Cre/βCat Flox(ex3)/+ mice in which stabilized β-catenin lacking the GSK3β phosphorylation sites becomes prematurely active in all basal keratinocytes during embryogenesis. In these mice, the IFE and SG adopt a HF fate (hair shaft; Kretzschmar *et al*, 2016). We observed induced ectopic Sox9 expression in K14Cre/βCat Flox(ex3)/+ epidermis. Nevertheless, premature and broad epidermal β-catenin activation, even in this developmental context, did not trigger Gata6 expression (Fig 2D).

We conclude that although Gata6 is a mutant Lef1 target gene, Gata6 expression is neither Lef1 nor β-catenin-dependent during pilosebaceous unit morphogenesis.

### The Gata6 lineage participates in sebaceous gland morphogenesis

We next investigated the role of Gata6 in SG morphogenesis. We did not observe any major SG abnormalities when Gata6 was deleted via the Krt5 promoter (cKO; Donati *et al*, 2017). However, an in-depth analysis of tail epidermis did indicate that in the absence of Gata6 the proportion of hypotrophic SG was significantly higher than in WT and heterozygous mice, albeit constituting a minority of the total SG (Fig 3A).

By overexpressing Gata6 in primary murine keratinocytes, we previously demonstrated that Gata6 induces a SD/SG transcriptional program upon differentiation *in vitro* (Donati *et al*, 2017). In addition to SD genes, Gata6 triggered the expression of genes associated with sebocyte differentiation, such as Pparg and Fasn, and downregulated genes associated with the androgen receptor gene signature that is a distinctive feature of the base of the SG (Donati *et al*, 2017). To overexpress Gata6 in developing epidermis, we performed *in utero* lentiviral infection (Beronja *et al*, 2013) with a Gata6-ires-GFP virus or a control virus expressing GFP alone (Figs 3B and C). This protocol is characterized by a low infection efficiency, with most transduced cells carrying only one transgene to avoid uncontrolled overexpression and off-target effects (Beronja *et al*, 2013). Gata6 overexpression in epidermal cells at E9.5 recapitulated aspects of the K14ΔNLef1 phenotype, in particular the formation of epithelial cysts with ectopic sebocytes (Fig 3B). Gata6 overexpression also resulted in the ectopic expression of the differentiating

**Figure 2.  Gata6 expression is independent of Wnt/β-catenin signaling.**

A    Sections of WT embryonic skin at different HF stages stained for Gata6, Lef1 or Sox9, and counterstained with Dapi. Black asterisks indicate overexposed areas of nonspecific Lef1 staining in the suprabasal epidermis. Quantification of the percentage of cells labeled for both Gata6 and Lef1, or Gata6 and Sox9 in stage 4–5 HF is shown (upper right panel). Data are means ± SD and were obtained from 9 HF from 3 mice.

B    Sections of E18.5 WT and Lef1$^{-/-}$ mouse skin stained for Lef1 and Gata6. Deletion of Lef1 does not impair Gata6 expression.

C    Sections of WT and K14ΔNβ-CateninER (K14ΔNβ-CatER D2 and D4 strains) adult dorsal skin stained with antibodies against Krt14 and Gata6. Topical treatment with 1 or 6 doses of 4OHT activates β-catenin, leading to anagen induction and ectopic hair follicles but not Gata6 expression.

D    Sections of E18.5 WT and K14Cre/βCat Flox(ex3)/+ mouse skin stained for Sox9 and Gata6. Activation of Wnt/β-catenin signaling during epidermis development does not induce Gata6 expression but results in ectopic expression of the HFSC marker Sox9.

Data information: (A–C) Scale bars: 50 μm. (D) Scale bar: 25 μm.

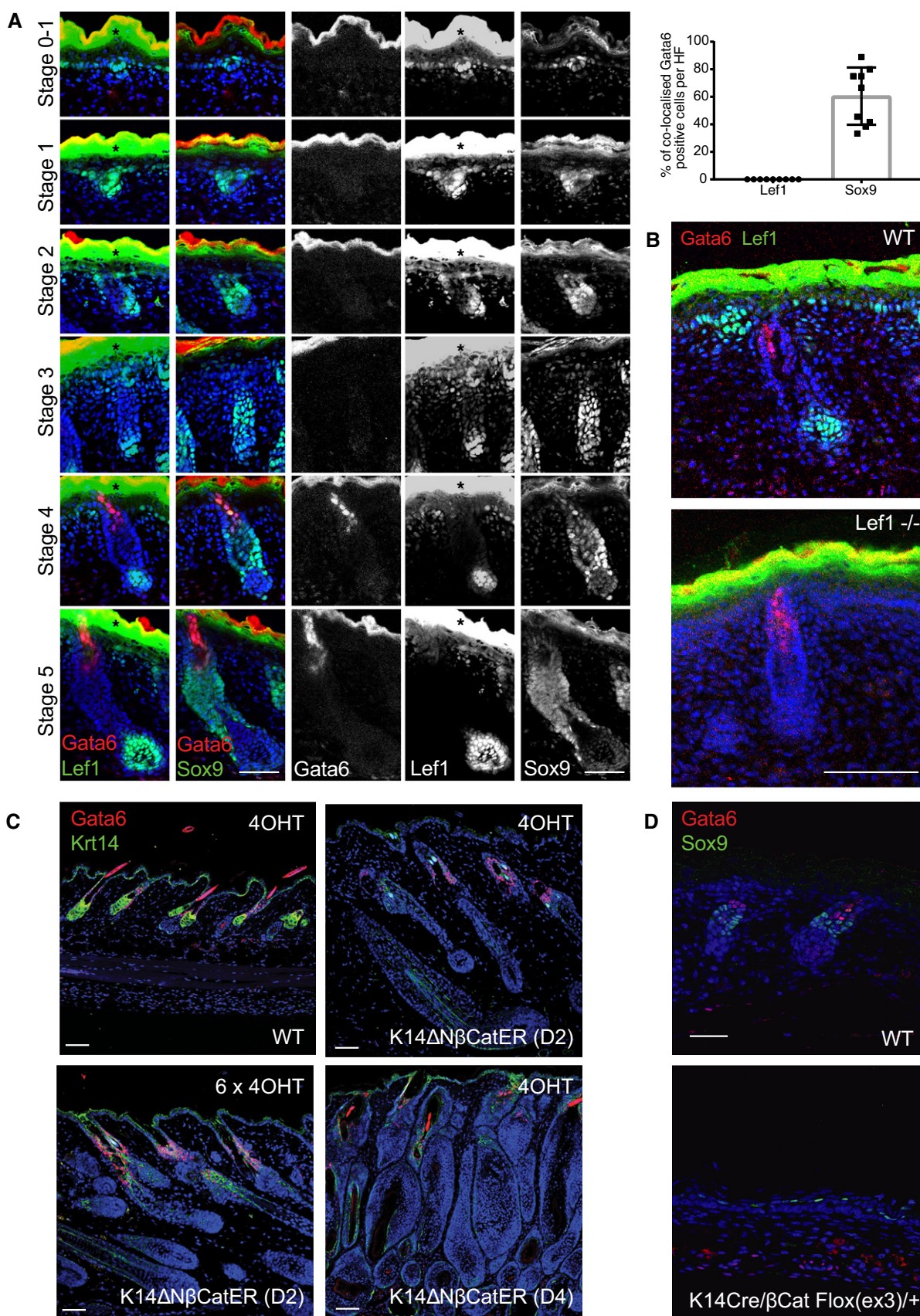

**Figure 2.**

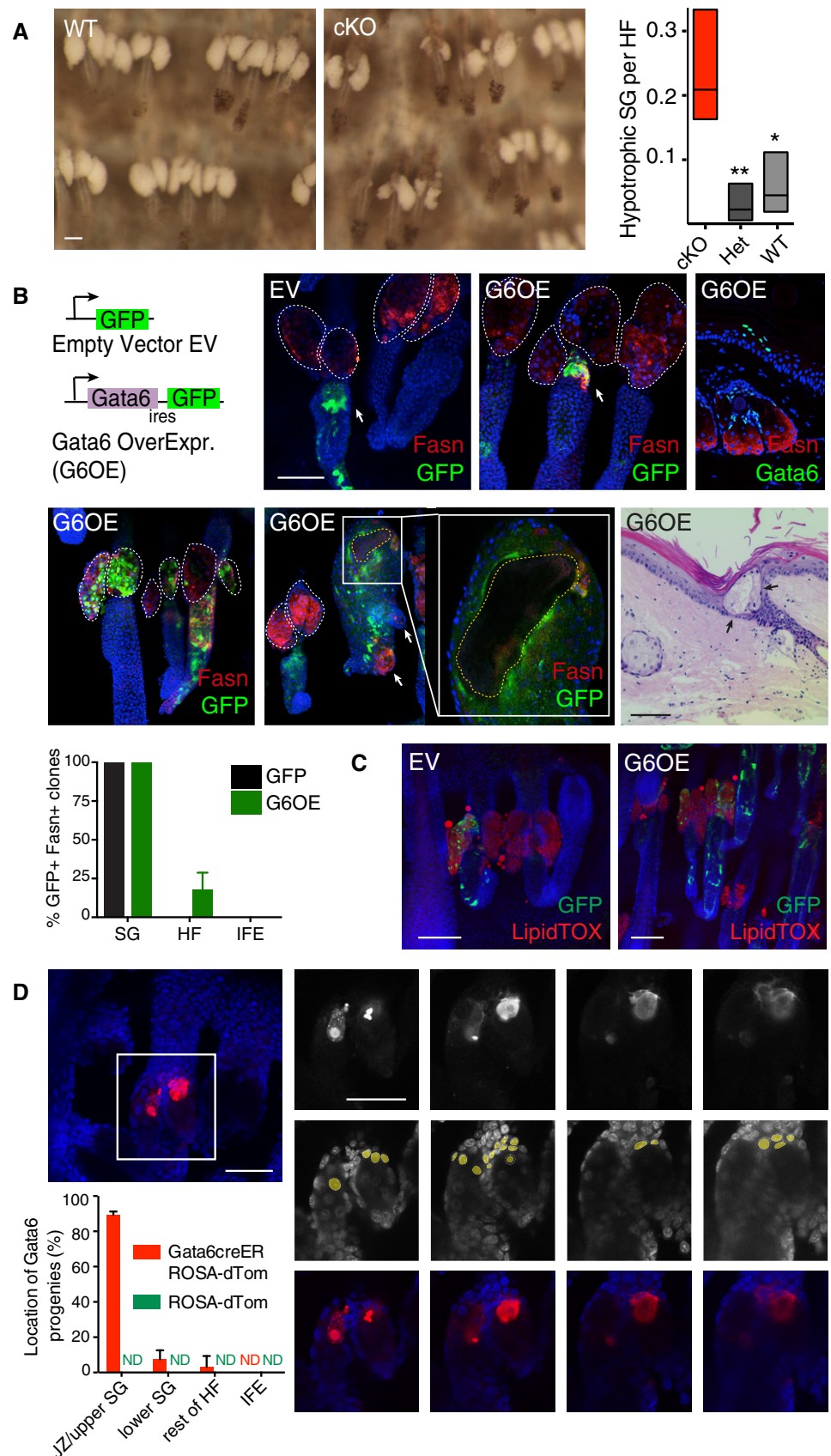

**Figure 3.**

**Figure 3.  Gata6 progenitors contribute to SG morphogenesis.**

A   Bright-field images of whole-mount WT and Gata6 cKO epidermis, showing abnormal SG upon loss of Gata6. Quantification of hypotrophic SG in WT, heterozygous (Het), and Gata6 cKO epidermal whole mounts. Data are boxplots with indication of means. Box limits are minimum and maximum values. An average of 165 HF per mouse (from 3 to 4 mice per genotype) was analyzed. *P < 0.05; **P < 0.005; unpaired Student's *t*-test.

B   Schematic representation of plasmid constructs used for *in utero* lentiviral infection. Whole mounts or sections of adult infected epidermis with empty vector (EV) or Gata6-ires-GFP (G6OE) lentivirus were stained for GFP and Fasn. *In vivo* overexpression of Gata6 leads to ectopic Fasn expression in the HF/SG unit. White dotted lines define SG, and yellow dotted lines define a cyst. Note that the cyst is mostly negative for Fasn in agreement with its SD-like phenotype. White arrows indicate GFP-positive infected cells. These cells are stained with Fasn only on G6OE expression. H&E-stained skin from G6OE mice shows a cyst with SG elements in the HF unit (black arrows). Staining for Gata6 (both endogenous and exogenous) shows that Gata6 expression occurs in a limited number of cells (representative image in upper right panel). Bottom left graph shows quantification of the percentage of clones labeled for both Fasn and GFP in the SG, HF, and IFE compartments. Data are means ± SD and originate from 4 EV mice and 8 G6OE mice (average of 11 clones per mouse).

C   Whole-mount adult epidermis infected with EV or G6OE lentivirus stained with LipidTOX. Ectopic Gata6 expressing cells are not stained with LipidTOX, indicating incomplete sebaceous maturation.

D   Lineage tracing experiments in Gata6EGFPCreERT2:Rosa26-fl/STOP/fl-tdTomato (Gata6creER ROSA-dTom) mice. A single dose of 4OHT was injected into pregnant females at E16.5. Tail skin from pups was collected at P13. Representative example of whole-mount epidermis showing tdTomato-labeled cells counterstained with Dapi (top left panel). Right panels show the different Z-stacks related to this whole mount. Gata6 progeny are mainly found in the upper SG/JZ. Localization of Gata6 progeny (dTomato+) was quantified in 20–26 pilosebaceous units per mouse (N = 4 mice) (bottom left panel). Data are means ± SD. As a control, quantification was also performed on Rosa26-fl/STOP/fl-tdTomato (ROSA-dTom) mice that were treated similarly to Gata6creER ROSA-dTomato mice.

Data information: (A–C) Scale bars: 50 μm. (D) Scale bar: 25 μm.

sebocyte marker Fasn in the SG and in the HF but not in the IFE, suggesting that Gata6 is able to promote SG identity in a cell compartment-dependent manner (Fig 3B). Gata6-induced ectopic sebocytes did not complete their maturation, as shown by the absence of LipidTOX staining (Fig 3C). This is consistent with the absence of Gata6 staining in differentiated sebocytes in mouse skin (Fig EV2B; Donati *et al*, 2017).

The origins of the SG during development are still a subject of debate (Reuter & Niemann, 2015). Primitive sebocytes can be detected in stage 5 HF (bulbous hair peg) as described by Paus *et al* (1999). While it is generally assumed that the entire gland is derived from a single lineage expressing Sox9 (Nowak *et al*, 2008; Frances & Niemann, 2012), lineage tracing via retrovirus-mediated LacZ transduction of dermabraded skin has shown that individual SG can be derived from more than one cell population (Ghazizadeh & Taichman, 2001). To examine this without having to damage the skin or rely on candidate SG markers, we generated WT:GFP chimeric mice (Fig EV3A) as previously described (Arwert *et al*, 2010). Aggregation chimeras are powerful tools that have been extensively used to infer developmental mechanisms (Tam, 2003). We assessed GFP expression in SG from 5 chimeric mice. In labeled SG, GFP staining was frequently localized exclusively to the upper SG (Fig EV3A). Therefore, we postulate that the SG is formed from at least two progenitors, one of which exclusively populates the upper SG and ducts.

To test whether Gata6 progeny were involved in morphogenesis of the upper SG, we performed lineage tracing by inducing recombination in Gata6EGFPCreERT2 (Gata6creER) × Rosa26-fl/STOP/ fl-tdTomato (ROSA-dTom) mice at E16.5 and E18.5 and analyzing whole mounts of tail and back skin 2 days and 17 days (P13) postlabeling (Figs 3D and EV3B and C). In line with our Gata6 expression analysis (Fig 2A), only cells in the upper part of stage 4–5 HFs were labeled 2 days after recombination (Fig EV3B). HF labeling efficiency was higher at E18.5 due to the higher abundance of stage 4–5 HF, but Gata6 progeny were never observed in earlier HF stages (Fig EV3B). At P13, Gata6 progeny were specifically located in the upper SG and JZ in approximately 90% of labeled pilosebaceous units (Fig 3D). Gata6 progeny were only rarely found in the lower SG or in the lower part of the HF (Fig EV3C).

Taken together, these data suggest that the Gata6 lineage is responsible for generating the upper part of the SG, including the SD.

## Gata6 specifies number and type of K14ΔNLef1 tumors

We next investigated the role of Gata6 in epidermal carcinogenesis. Unlike WT mice, K14ΔNLef1 mice develop tumors spontaneously, or after a single DMBA application (Niemann *et al*, 2007). In both contexts, K14ΔNLef1 tumors have elements of sebaceous differentiation. Gata6 was expressed in all K14ΔNLef1 tumors, whereas it was undetectable in DMBA/TPA-induced papillomas in WT mice (Fig 4A). Ablation of Gata6 in K14ΔNLef1 mice (K14ΔNLef1:cKO) resulted in an increased rate of tumor formation following DMBA treatment, and in an increased number of tumors per mouse (Fig 4B), indicating that Gata6 acted as a tumor suppressor. In addition to increasing the overall number of tumors per mouse, loss of Gata6 led to an increase in papilloma-like tumors and a decrease in tumors with SG elements (Fig 4C). K14ΔNLef1 tumors strongly expressed the SD markers Plet1 and ATP6v1c2 (Donati *et al*, 2017), while WT and K14ΔNLef1:cKO tumors did not, indicating that SG differentiation was significantly reduced upon Gata6 loss (Fig 4D and Appendix Fig S1).

## Gata6 acts as a tumor suppressor by controlling the DNA mismatch repair response

Muir–Torre syndrome is a rare genetic condition that predisposes patients to developing sebaceous skin tumors and visceral malignancies. This autosomal dominant variant of Lynch syndrome is caused by mutations in DNA mismatch repair (MMR) genes, resulting in microsatellite instability (Eisen & Michael, 2009a,b; John & Schwartz, 2016). Since loss of Gata6 in cultured mouse keratinocytes leads to DNA damage, triggering apoptosis (Wang *et al*, 2017), we investigated whether the tumor suppressive function of Gata6 might be due to an effect on MMR gene transcription.

Computational analysis of RNA-Seq data from Gata6-deficient keratinocytes (Wang *et al*, 2017) revealed that DNA metabolism

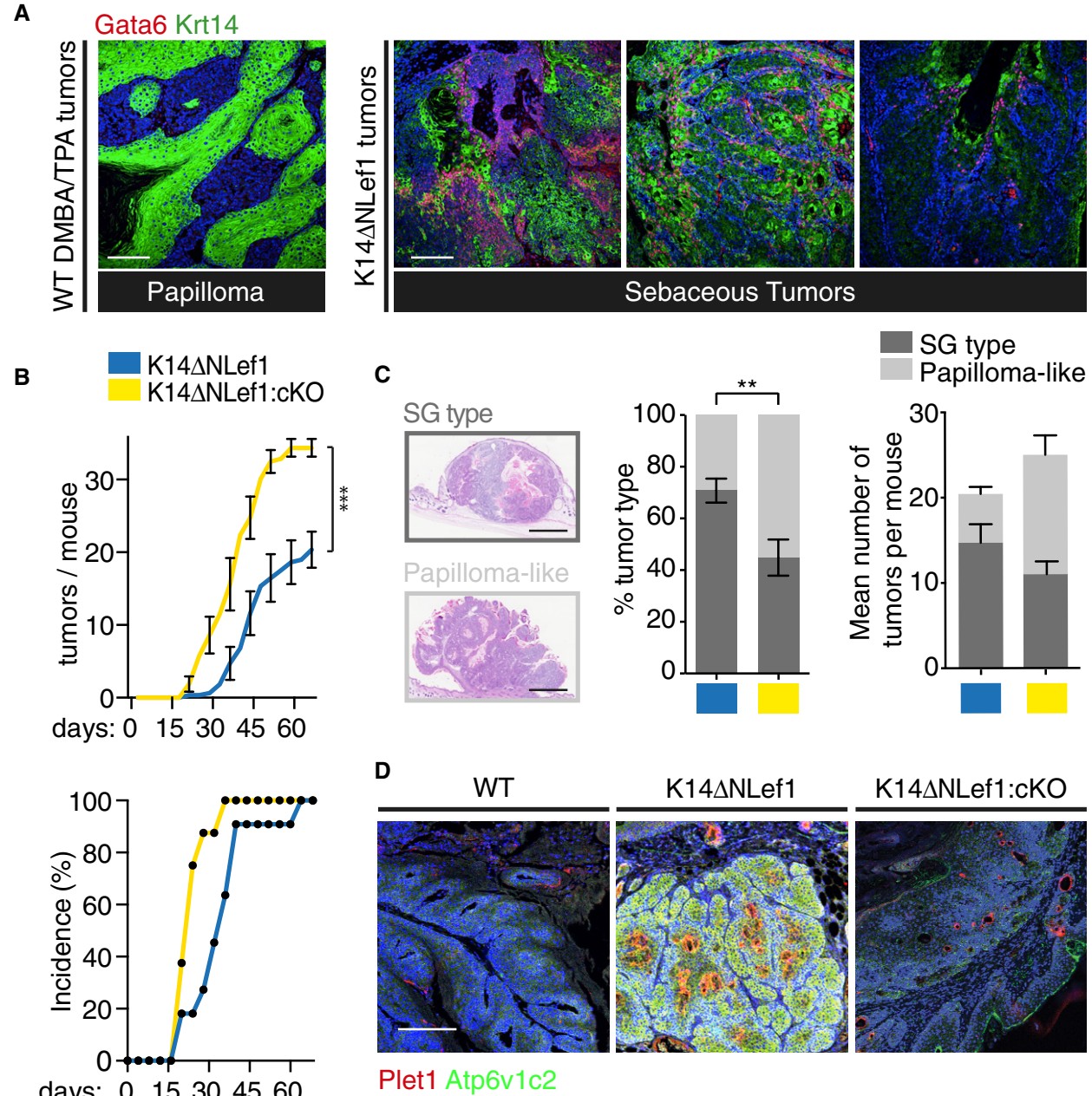

**Figure 4.   Effect of Gata6 loss on K14ΔNLef1 sebaceous tumors.**

A   Sections of mouse skin tumors were stained with antibodies to Gata6 and Krt14. A papilloma from a WT mouse treated with DMBA and TPA (left panel) is compared with sebaceous tumors found in K14ΔNLef1 mice (right panels).

B   Tumor burden and tumor incidence in DMBA-treated K14ΔNLef1 (*n* = 11) and K14ΔNLef1:cKO (*n* = 8) mice. ***P* < 0.001; Wilcoxon matched-pairs signed rank test.

C   Representative H&E-stained tumors of K14ΔNLef1 and K14ΔNLef1:cKO mice with quantification of each tumor type relative to the total number of tumors in each group (middle panel) and as the average absolute number of tumors per mouse (right panel). A total of 268 tumors were analyzed. 143 tumors were found in 7 K14ΔNLef1 mice (103 sebaceous tumors and 40 papillomas), while 125 were found in 5 K14ΔNLef1:cKO mice (55 sebaceous tumors and 70 papillomas). ***P* < 0.005, Student's *t*-test.

D   Representative images of mouse skin tumor sections labeled with antibodies to Plet1 and Atp6v1c2.

Data information: (A, D) Scale bars: 100 μm. (C) Scale bar: 1 mm. (B, C) Data are means ± SEM.

genes (including a number of genes related to DNA repair and replication) were over-represented in downregulated genes (Fig 5A and B). When we intersected Gata6 ChIP-Seq data (Donati *et al*, 2017) with the RNA-Seq data from WT and Gata6 cKO keratinocytes (Wang *et al*, 2017), we confirmed that KEGG pathways related to DNA repair, including MMR, were enriched

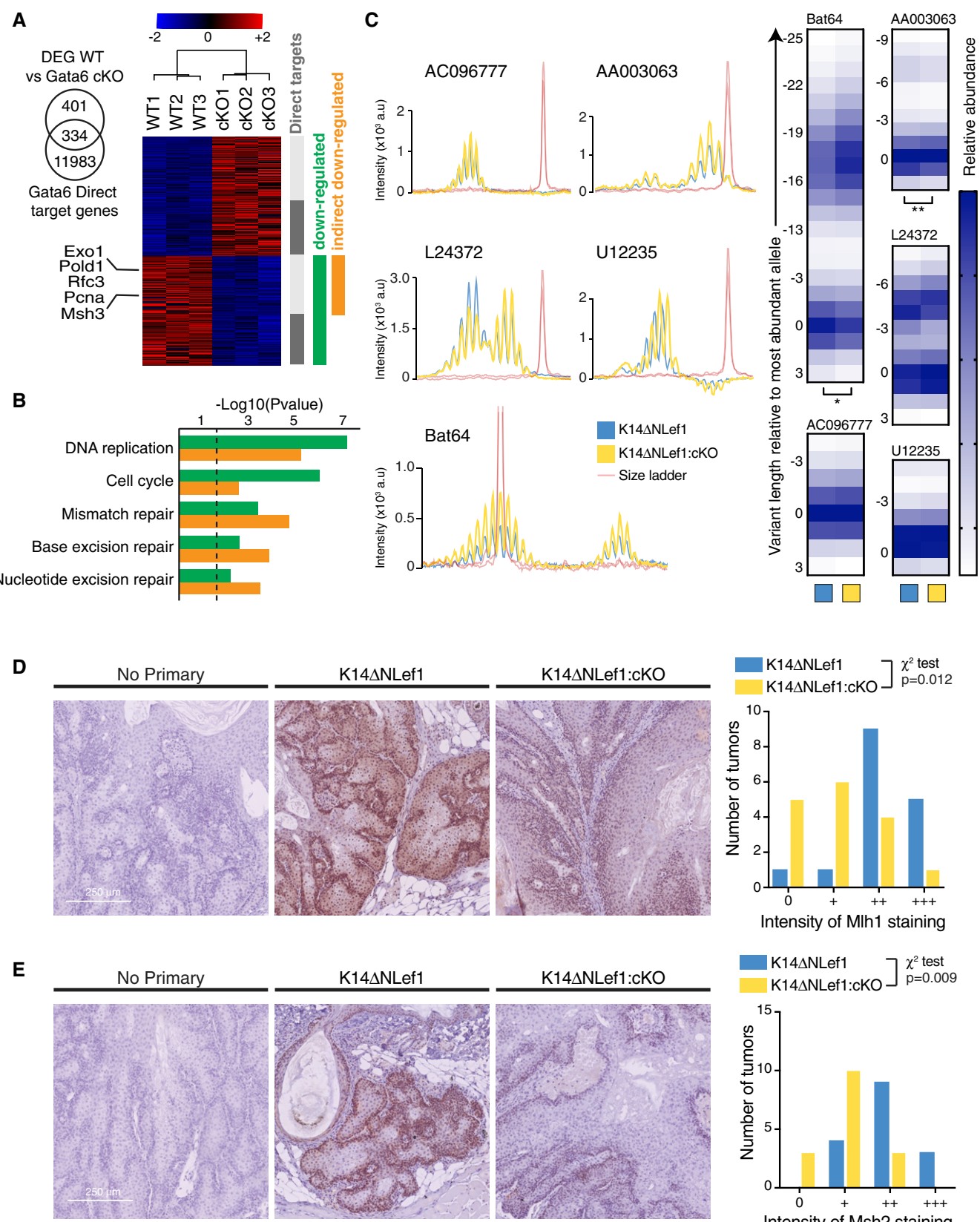

**Figure 5.**

◄

**Figure 5.  Gata6 controls the DNA mismatch repair response during sebaceous tumorigenesis.**

A   Heat-map highlighting differentially expressed genes (DEG) between WT and Gata6 cKO keratinocytes from Wang et al (2017). Gata6 direct transcriptional targets from Chip-Seq data (Donati et al, 2017) are shown in dark gray.
B   KEGG pathway analysis performed on downregulated genes upon Gata6 loss.
C   MSI analysis performed on skin tumors from DMBA-treated K14ΔNLef1 and K14ΔNLef1:cKO mice (5 mice in each group). Representative microsatellite profiles of K14ΔNLef1 and K14ΔNLef1:cKO tumors. Peak heights were normalized to the highest peak in each microsatellite profile to obtain the relative frequency of each allele for five different markers. "0" indicates the position of the highest peak in bp. *$P < 0.05$; **$P < 0.001$; paired t-test.
D   Representative immunohistochemistry staining for Mlh1 performed on skin tumors from DMBA-treated K14ΔNLef1 and K14ΔNLef1:cKO mice. Four mice were included in each group. A technical control is displayed (without primary antibody incubation). Semi-quantitative analysis of Mlh1 staining and statistical analysis (chi-square test) were performed.
E   Representative immunohistochemistry staining for Msh2 and semi-quantitative analysis performed as in (D).

Data information: (D and E) Scale bar: 250 μm.

in the genes downregulated on loss of Gata6 (Fig 5A and B). The specific MMR genes Msh3, Exo1, Pold1, Rfc3, and Pcna were downregulated in Gata6 cKO keratinocytes (Fig 5A), although they were not identified as direct transcriptional targets of Gata6 (Donati et al, 2017).

Using ChIP-Atlas, an extensive database of publicly available ChIP-Seq experiments (Oki et al, 2018 and http://chip-atlas.org), we analyzed Gata6 ChIP-Seq experiments performed in human ES cell-derived mesendodermal cells and in human colon, gastric and pancreatic adenocarcinoma cell lines. We identified eight genes in the MMR pathway for which a Gata6-binding peak was found within 10 kb from the transcription start in at least 2 different datasets: Msh3, Rfc2, Pold4, Ssbp1, Pold3, Exo1, Rfc1, and Mlh3 (Fig EV4). It is possible that these were not found in Gata6-overexpressing primary mouse keratinocytes (Donati et al, 2017) because the ChIP-Seq analyses were performed too soon after lentiviral infection for Gata6 to induce MMR genes. Nevertheless, the MMR genes identified as Gata6 direct transcriptional targets using ChIP-Atlas closely match the MMR genes that were downregulated in Gata6 cKO keratinocytes and confirm that the MMR pathway is regulated by Gata6.

To further study the functional impact of Gata6 on the MMR pathway, we used two methods that are key to diagnosing Muir–Torre syndrome (Eisen & Michael, 2009b) and Lynch syndrome (Giardiello et al, 2014): microsatellite instability (MSI) testing and immunohistochemistry for MMR proteins. We first evaluated MSI in skin tumors from K14ΔNLef1 mice and K14ΔNLef1:cKO mice. We analyzed the relative allele frequency of five microsatellite regions, as previously described (Woerner et al, 2015; Germano et al, 2017; Keysselt et al, 2017). Three markers (Bat64, AA003063, and L24372) showed a shift in allele distribution, two of which were

significant (Fig 5C). This indicated a more unstable microsatellite phenotype in K14ΔNLef1:cKO mice.

The MMR pathway corrects errors within newly synthesized DNA strands during replication and mainly relies on MutSα and MutLα complexes formed by Msh2/Msh6 and Mlh1/Pms2, respectively (Jiricny, 2006). Mlh1 (Fig 5D and Appendix Fig S2A) and Msh2 (Fig 5E and Appendix Fig S2B) expression were significantly reduced in K14ΔNLef1:cKO as compared to K14ΔNLef1 tumors. The broad downregulation of MMR genes, in particular of Msh3, in Gata6-deleted skin is likely to explain the downregulation of Mlh1 and Msh2 (Figs 5D and E, and Appendix Fig S2) through destabilization of MMR protein complexes.

Altogether, these results indicate that the increased incidence of tumors in K14ΔNLef1:cKO mice could be due to a decrease in expression of Gata6-dependent MMR proteins.

## Gata6 expression is a hallmark of human skin tumors with sebaceous differentiation

To examine whether the observations in mice were relevant to human skin tumors, we first confirmed that Gata6 was expressed in the same locations in human as in mouse adult skin. Indeed, Gata6 was expressed in the upper SG, SD, and JZ (Fig 6A). We then analyzed Gata6 expression in a large panel of benign and malignant human skin samples ($N = 73$; Fig 6B and Appendix Fig S3). Eighteen out of 19 human sebaceous tumors were Gata6-positive, regardless of their malignancy grade. In addition, the majority of the tumors harboring elements of SG differentiation were positive for Gata6: sebaceous nevus, syringocystadenoma papilliferum (SCAP, commonly associated with sebaceous nevus), folliculosebaceous cystic hamartoma (FSCH), steatocystoma multiplex (genetic

►

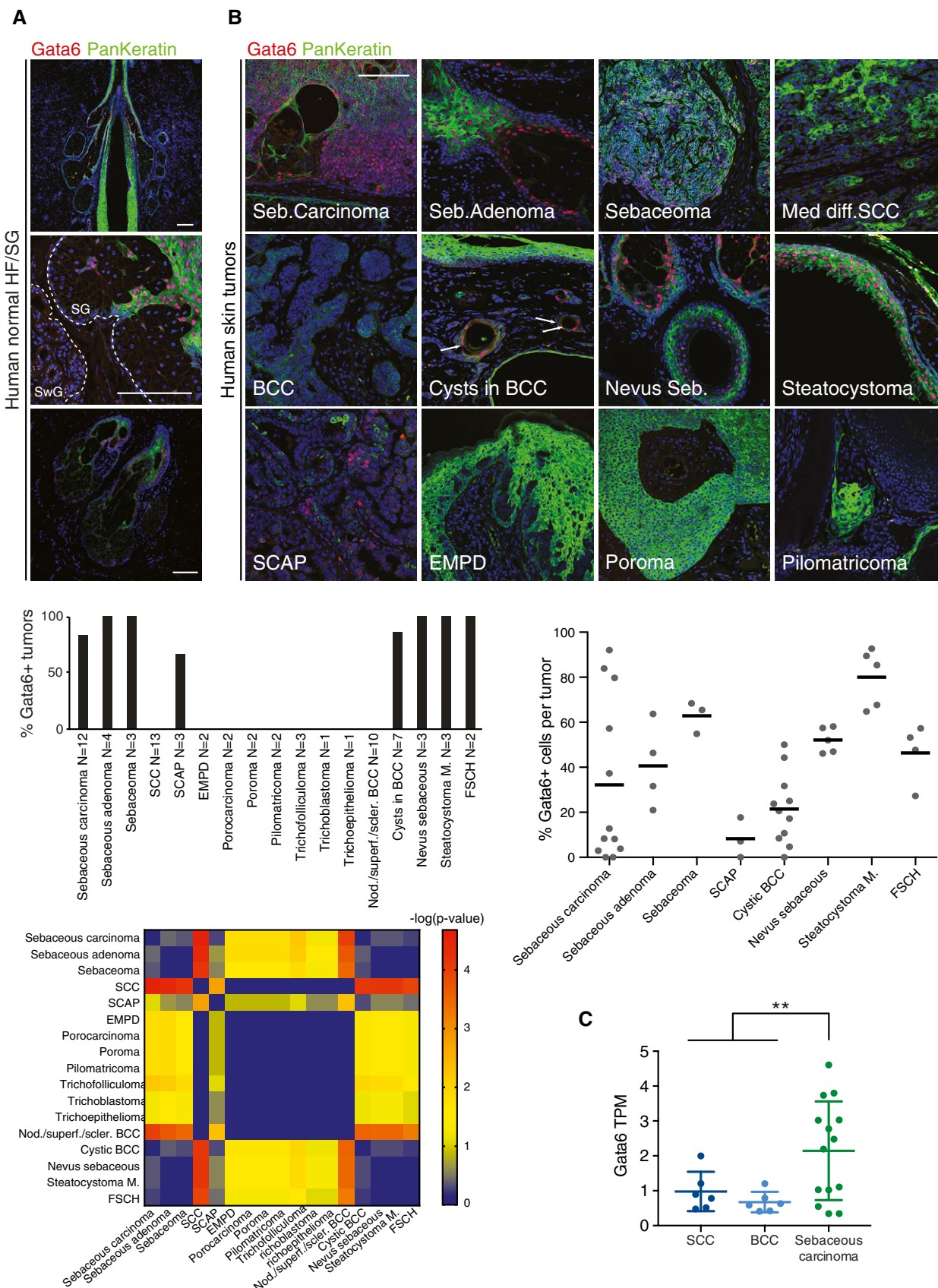

**Figure 6.**

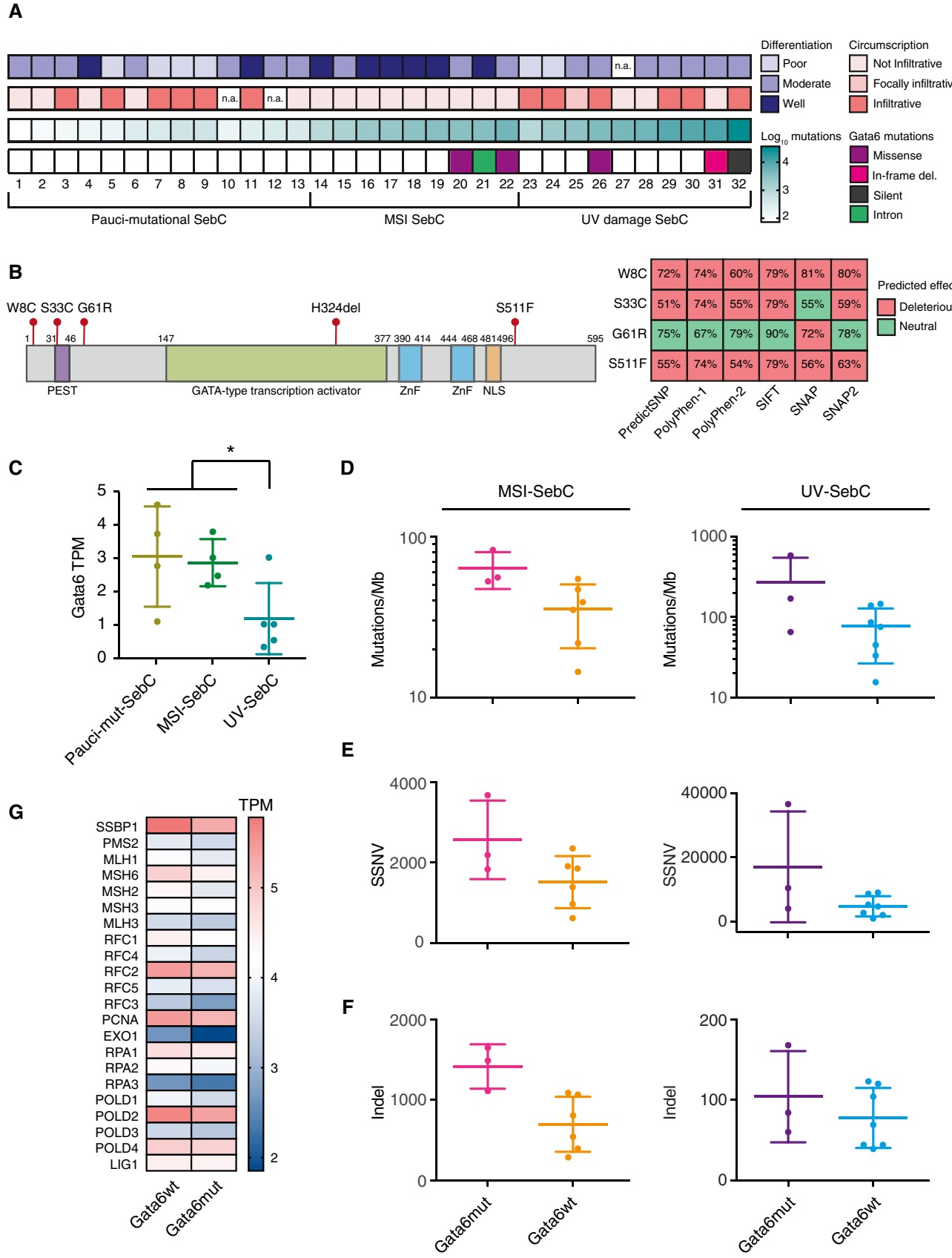

**Figure 7.**

◄

**Figure 7. Gata6 is mutated or downregulated in human sebaceous carcinomas with a high mutational burden.**

A    Differentiation status (poor, moderate, or well-differentiated), circumscription (not infiltrative, focally infiltrative, or infiltrative), $\log_{10}$ total number of mutations, and Gata6 mutations in 32 human sebaceous carcinomas (SebC) separated into 3 subgroups: pauci-mutational, MSI-related, and UV-related, as published by North *et al* (2018). For Gata6 mutations, the most deleterious mutation is displayed. n.a.: not assessed.

B    Schematic representation of Gata6 protein displaying the different protein domains: PEST, GATA-type transcription activator, zinc-finger (ZnF), nuclear localization sequence (NLS), and the different Gata6 missense mutations and deletions found in human SebC. Heat-map displaying the predicted effect (deleterious or neutral) of W8C, S33C, G61R, and S511F point mutations as assessed by PredictSNP, Polyphen-1, Polyphen-2, SIFT, SNAP, and SNAP2 algorithms (right panel). Percentages indicate the level of confidence of the predictions.

C    Gata6 expression (measured in transcripts per million, TPM) in human pauci-mutational (pauci-mut) SebC (*n* = 4), MSI-related SebC (*n* = 4), and UV-related SebC (*n* = 5) samples. Data are means ± SD. *$P \leq 0.05$, Student's *t*-test.

D–F  Graphs showing the number of mutations per megabase (mutations/Mb) (D), the number of somatic single-nucleotide variants (SSNV) (E), and Indel (F) of Gata6mut (*n* = 3) or Gata6wt (*n* = 6) MSI-related SebC (left panels), as well as of Gata6mut (*n* = 3) or Gata6wt (*n* = 7) UV-related SebC (right panels). Data are means ± SD.

G    Heat-map displaying the mean expression level of MMR genes (measured in TPM) in Gata6wt (*n* = 10) and Gata6mut (*n* = 3) SebC.

condition characterized by multiple benign sebaceous cysts), and basal cell carcinoma (BCC) with a cystic differentiation. In contrast, squamous cell carcinomas (SCC), HF/matrical tumors, and extra-mammary Paget's disease (EMPD) were negative for Gata6 (Fig 6B and Appendix Fig S3). The percentage of Gata6-positive cells was typically in the same range within each tumor group, except in the case of sebaceous carcinomas, which exhibited more variability in Gata6 expression (Fig 6B and Appendix Fig S3).

Two BCC subtypes could be distinguished on the basis of Gata6 expression. Gata6 was selectively expressed in BCC with cysts. We speculate that these BCC may arise from a different region of the HF compared to cyst-less BCC. In addition, although the number of samples of the benign skin disorder SCAP was limited, we observed Gata6 expression in almost 70% of cases (Fig 6B and Appendix Fig S3). This is intriguing because SCAP is thought to derive from sweat glands (Yamamoto *et al*, 2002), yet normal sweat glands do not express Gata6 (Fig 6A).

In support of our findings, we reanalyzed published RNA-Seq data from human SCC, BCC, and sebaceous carcinomas (SebC; North *et al*, 2018). We found that Gata6 expression was significantly increased in SebC as compared to BCC and SCC (Fig 6C), which confirms our previous observations (Fig 6B and Appendix Fig S3).

Therefore, our results establish Gata6 as a key histological marker of human skin tumors that originate from, or have differentiated elements of, the SG.

**Mutation and downregulation of Gata6 are features of sebaceous carcinomas with a high mutational burden**

North *et al* (2018) have distinguished three subclasses of human SebC based on their mutational profile: the ocular and cutaneous pauci-mutational SebC (with a low prevalence of mutations); SebC with a MSI mutational signature (intermediate prevalence of mutations) and SebC with a UV mutational signature (highest somatic mutation burden). Within this dataset, we found GATA6 mutations in approximately 30% of SebC harboring a MSI or UV damage signature (Fig 7A). GATA6 missense mutations were mostly deleterious (Fig 7B). In addition, analysis of RNA-Seq data showed that Gata6 expression was significantly lower in UV-induced SebC than in other SebC (Fig 7C). In agreement with our observations in mice (Fig 4B and C), UV-related SebC expressing a low level of Gata6 were less differentiated and more aggressive than the pauci-mutational and MSI-mutant tumors (North *et al*, 2018).

Despite the different mutational mechanisms associated with MSI-related and UV-induced SebC, Gata6-mutated tumors (Gata6-mut) displayed a higher number of mutations/Mb (Fig 7D), of somatic single-nucleotide variants (SSNV) (Fig 7E) and Indel (Fig 7F) than Gata6 wild-type tumors (Gata6wt). In addition, Gata6-mut tumors displayed a trend of downregulation in MMR genes when compared to Gata6wt tumors (Fig 7G). The limited number of Gata6mut SebC samples did not allow us to test for significance. However, these results suggest that Gata6 affects DNA damage pathways in human sebaceous tumors as in mice.

Our data indicate a role of Gata6 in the physiopathology of sebaceous tumors, particularly in relation to DNA mismatch repair.

## Discussion

Gata6 is widely expressed in the heart and in endoderm-derived tissues, including the lungs, liver, pancreas, stomach, and intestine (Maeda *et al*, 2005). Mice lacking Gata6 die during embryonic development as a result of endoderm defects (Morrisey *et al*, 1998). In contrast, Gata6 expression in the epidermis is limited to a population of SD/SG progenitors in the developing HF and to the JZ, upper SG, and part of the infundibulum in adult skin (Figs 2A and 6A, and EV2A–C; Donati *et al*, 2017).

Although Gata6 is strongly upregulated in the epidermis of K14ΔNLef1 mice (Fig 1H) and is a direct ΔNLef1 target gene (Fig 1C), we saw no evidence for co-expression of Gata6 and Lef1 in developing or adult epidermis and Gata6 was not induced upon β-catenin activation (Fig 2), even though Gata6 synergizes with or activates Wnt signaling in a number of contexts (Afouda *et al*, 2008; Zhang *et al*, 2008b; Whissell *et al*, 2014).

During skin morphogenesis, Gata6 was first expressed in the upper stage 4 HF (Fig 2A), consistent with a recent study that reported co-localization with Krt79 and Lrig1 (Mesler *et al*, 2017). The Gata6-positive population does not originate from Shh-positive progenitors (Mesler *et al*, 2017) and an alternative possibility is that it arises from Sox9 progenitors (Fig 2A). Regulation of Sox9 by Gata6 has been shown in several organs, including the pancreas (Carrasco *et al*, 2012) and cardiac valves (Gharibeh *et al*, 2018).

Gata6 gain and loss-of-function experiments revealed that Gata6 regulates JZ, SD, and upper SG differentiation in WT mice (Figs 3 and EV3B and C). In K14ΔNLef1 mice, a ΔNLef1-Gata6

transcriptional cascade triggers formation of multilayered cysts and ectopic SG (Fig 1A and B). Most of the cysts represent aberrant SD and JZ, since they express SD/JZ markers (Donati *et al*, 2017), rather than IFE as previously proposed (Niemann *et al*, 2002). Gata6 overexpression *in utero* led to formation of cysts, reminiscent of ducts, with ectopic SG features (Fig 3B and C). Our functional data together with our lineage tracing experiments (Figs 3D and EV3B and C) allow us to propose that the Gata6-positive cells in the developing HF give rise to the upper SG. The lower SG has a distinct origin that has yet to be identified (Donati *et al*, 2017).

Gata6 is overexpressed in a variety of cancers, including pancreato-biliary cancers (Kwei *et al*, 2008), colon cancers (Shureiqi *et al*, 2007; Tsuji *et al*, 2014), esophageal adenocarcinomas (Lin *et al*, 2012), breast cancers (Song *et al*, 2015), and adrenal tumors (Vuorenoja *et al*, 2007). Conversely, it is lost in certain ovarian cancers (Cai *et al*, 2009) and may act as a tumor suppressor gene in lung cancer (Cheung *et al*, 2013) and astrocytoma (Kamnasaran *et al*, 2007). We observed a strong association between Gata6 and sebaceous carcinogenesis in mouse and human skin. Gata6 was expressed in sebaceous tumors of K14ΔNLef1 mice (Fig 4A) and deletion of Gata6 reduced the proportion of tumors with sebaceous differentiation (Fig 4C). Furthermore, human skin tumors with sebaceous differentiation expressed Gata6 (Fig 6B and C). Thus, Gata6 could be a useful biomarker for diagnostic pathology of sebaceous tumors.

We observed that Gata6-positive cells were Ki67-negative in WT mouse back skin (Fig EV2B). However, we previously showed in K14ΔNLef1 mice that cell proliferation is a downstream effect of ΔNLef1 overexpression (Niemann *et al*, 2002). Ki67 staining is found in the outer root sheath and in some cells at the periphery of the dermal cysts. In addition, Ki67-positive cells are twice as frequent in the basal layer of K14ΔNLef1 as WT IFE (Niemann *et al*, 2002). Thus, Ki67 is mostly expressed in the Gata6-negative regions of K14ΔNLef1 epidermis. This suggests that Gata6 does not stimulate proliferation, consistent with previous observations (Donati *et al*, 2017). Therefore, two cell populations can be distinguished: actively proliferating K14$^+$/ΔNLef1$^+$/Gata6$^-$ keratinocytes and infrequently proliferating K14$^+$/ΔNLef1$^+$/Gata6$^+$ keratinocytes of the JZ and upper SG. K14ΔNLef1 mice develop sebaceous tumors at high frequency. This suggests that tumor initiation involves cooperation between these two cell populations. Our data indicate that Gata6 is likely to be responsible for the sebaceous differentiation observed in the tumors (Fig 4C) but also acts as a tumor suppressor (Fig 4B). In addition, we and others have confirmed the existence of proliferative cells in K14ΔNLef1 sebaceous tumors by showing Ki67 expression (Niemann *et al*, 2002), BrdU incorporation (Niemann *et al*, 2007), and isolating tumor-propagating cells that form secondary tumors in serial transplantation assays (Petersson *et al*, 2015). How the two cell populations cooperate to initiate tumors remains to be elucidated.

Sebaceous tumors are closely associated with MMR deficiency, for example, in the context of Muir–Torre syndrome (Eisen & Michael, 2009a,b; John & Schwartz, 2016). In this syndrome, mutations in the MMR genes Msh2 or, less frequently, Mlh1 and Msh6, predispose cells to DNA base errors (shown by the acquisition of MSI status; John & Schwartz, 2016). The estimated frequency of MSI is at least 60% of all sebaceous neoplasms, while only 3% of SG hyperplasia (Kruse *et al*, 2003; Jessup *et al*, 2016). Furthermore,

Msh2-deficient mice develop sebaceous tumors (Reitmair *et al*, 1996).

We observed that Gata6 was required for expression of MMR genes. Microsatellite stability and expression of Msh2 and Mlh1 were reduced upon Gata6 knockout in K14ΔNLef1 mice (Figs 5C, D and E, and Appendix Fig S2). This would explain why tumor incidence was increased in K14ΔNLef1:cKO mice (Fig 4B). Paradoxically, while MSI-linked tumors develop and progress rapidly, their prognosis is often better. The increased mutation load resulting from MMR inactivation generates multiple neo-antigens and stimulates immune surveillance and cancer clearance (Germano *et al*, 2017). MMR-deficient tumors show an excellent response to pembrolizumab, an immunotherapy targeting the PD-1 receptor (Le *et al*, 2015). In addition to MMR regulation, Gata6 may also suppress tumor formation by reducing proliferation and increasing SD/SG differentiation (Donati *et al*, 2017).

Like K14ΔNLef1 mice, mice overexpressing ΔNLef1 under the control of the Krt15 promoter (which labels HF bulge cells) develop sebaceous tumors (Petersson *et al*, 2015). These tumors are more aggressive than those of K14ΔNLef1 mice and are associated with increased DNA damage (Petersson *et al*, 2015). These results could be explained by the lack of Gata6 expression in the bulge.

In humans, sebaceous tumors harbor a high level of expression of Gata6 (Fig 6B and C, and Appendix Fig S3). However, sebaceous carcinoma can also acquire Gata6 mutations correlating with the total mutational burden (Fig 7). Gata6mut tumors display reduced expression of several MMR genes (Fig 7G). In addition, a reduction in Gata6 expression is associated with less differentiated and more aggressive UV-related SebC, independent of GATA6 mutational status (Fig 7C).

The link between Gata6 and MMR that we have found could open up new therapeutic approaches to target sebaceous carcinogenesis. It is tempting to speculate that the MMR DNA repair pathway is necessary for maintenance of healthy SG, because lipid production is associated with the formation of reactive oxygen species (Bek-Thomsen *et al*, 2014; Ibrahim *et al*, 2014) and subsequent DNA damage.

## Materials and Methods

### Mice

All animal procedures were subject to local ethical approval and performed under a UK Government Home Office license (PPL 70/8474). K14ΔNLef1 (Niemann *et al*, 2002), Lef1$^{-/-}$ (Van Genderen *et al*, 1994), K14ΔNβ-CateninER (D2 and D4 lines; Lo Celso *et al*, 2004), K14Cre/βCat Flox(ex3)/+ (Zhang *et al*, 2008a), epidermal Gata6 conditional knockout (cKO) (Donati *et al*, 2017), Gata6EGFPCreERT2 (Donati *et al*, 2017), Lgr6EGFPCreERT2 (Snippert *et al*, 2010), and Rosa26-fl/STOP/fl-tdTomato (Madisen *et al*, 2010) mice were previously described. The K14ΔNβ-CateninER transgene was activated by one or six topical applications of 1.5 mg 4-hydroxytamoxifen (4OHT) (Sigma) (Donati *et al*, 2017). For lineage tracing experiments, pregnant females were injected intraperitoneally with a dose of 50 μg/g of tamoxifen (Sigma) at E16.5 and E18.5. Samples were collected at 2 or 17 days after recombination. Ultrasound-guided lentiviral *in utero* injection

procedures were performed as previously described (Beronja *et al*, 2013). Lentiviral particles were produced by transfecting HEK293 cells with a Trans-Lentiviral Packaging System in combination with Precision LentiORFs control and Gata6 (Dharmacon). GFP chimeric mice were generated as previously described (Arwert *et al*, 2010). For skin carcinogenesis experiments, K14ΔNLef1 and K14ΔNLef1:cKO mice received a single 100 nmol dose of DMBA (7,12-dimethylbenz(a)anthracene; Niemann *et al*, 2007). Tumor incidence and burden were assessed once a week by two independent researchers. As a control for Appendix Fig S2A and B, a WT mouse was UVB-irradiated with a total dose of 500 mJ/cm$^2$. No specific method for randomization, blinding, or estimation of sample size was used. Male and females were used. All efforts were made to minimize suffering of mice.

### Human tissue

All human tissue samples were collected, diagnosed, and processed for research in accordance with the recommendations of the relevant local ethics committees in compliance with the UK Human Tissue Act and approved by the National Research Ethics Service (08/H0306/30), German Medical Council, and/or the Japanese Ministry of Health, Labor, and Welfare. Human embryonic and fetal tissues were obtained with appropriate ethical approval from the UK Human Developmental Biology Resource (www.hdbr. org). Informed consent was obtained from all subjects. The experiments conformed to the principles set out in the WMA Declaration of Helsinki and the Department of Health and Human Services Belmont Report.

### Tissue processing and analysis

Sections from OCT- or paraffin-embedded tissues and epidermal whole mounts were prepared and processed as previously described (Donati *et al*, 2017; Walko *et al*, 2017). Primary antibodies were used at the indicated dilutions: Fabp5 (1:100, R&D Systems AF1476); Krt15 (1:1,000, LHK-15 clone, Abcam ab80522); Lef1 (1:100–500, C12A5 clone, Cell Signaling 2230 and 8490); Gata6 (1:100–1,000, D61E4 clone, Cell Signaling 5851 and 26452); Krt14 (1:1,000, LL002 clone, Abcam ab7800 and 1:1,000, Covance SIG-3476); Sox9 (1:100, R&D Systems AF3075 and 1:400, D8G8H clone, Cell Signaling 71273); Ki67 (1:50, Tec3 clone, Dako); Lrig1 (1:200, R&D Systems AF3688); Cd34 (1:100, RAM34 clone, BD Biosciences 553731); Tcf3/4 (1:100, Abcam ab12065); pankeratin (1:1,000, clone LP34, LSBio LS-C95318); Fasn (1:100, G-11, Santa Cruz sc-48357); GFP (1:200, Abcam 6673 and 1:800, Thermo Fisher Scientific A-11122); Plet1 (1:200, LSBio LS-C149191); Atp6v1c2 (1:200, Sigma HPA034735); Mlh1 (1:100, Abcam ab92312); and Msh2 (1:100, Clone D24B5, Cell Signaling 2017). Dapi (Thermo Fisher Scientific) or hematoxylin (Dako) were used as counterstains. Where indicated, HCS LipidTOX Deep Red Neutral lipid stain was used (1:500, Life Technologies). Alexa Fluor-conjugated secondary antibodies (Thermo Fisher Scientific) were used for immunofluorescence and Vectastain ABC HRP Kit with Vector DAB or NovaRED HRP substrate kit (Vector Laboratories) for immunohistochemistry. Hematoxylin and eosin (H&E) staining was performed where indicated. Images were acquired with a Leica TCS SP5 Tandem Scanner confocal, a Nikon A1 confocal microscope, or a

NanoZoomer Slide Scanner (Hamamatsu). Digital images were processed using NIS-Elements Advanced Research (Nikon), ImageJ (https://imagej.nih.gov/ij/), or NDP.view2 (Hamamatsu).

### Flow cytometry

Epidermal cells were isolated, labeled with anti-CD34 (RAM34 clone, BD Biosciences 553731) and anti-CD49f (Itga6) (GoH3 clone, eBioscience 14-0495-81), and sorted by flow cytometry as previously described (Jensen *et al*, 2010). Data were analyzed using FlowJo software.

### ChIP-Seq, microarray, and computational analysis

ChIP-Seq was performed as previously described (Mulder *et al*, 2012). Briefly, crosslinked material corresponding to ∼$10^7$ K14ΔNLef1 cultured primary keratinocytes (collected from 9.5-week-old mice) was incubated overnight with 10 μg of Lef1 (Cell Signaling) or Flag antibody (Sigma), and immune-precipitated DNA was sequenced. Base-calling, genome alignment, filtering against potential PCR duplications and peak calling were performed as previously described (Donati *et al*, 2017).

RNAs from FACS-purified Cd34$^+$Itga6$^+$ and Cd34$^-$Itga6$^+$ epidermal cells collected from WT and K14ΔNLef1 epidermis (9.5-week-old mice) were provided to the Paterson Institute Microarray Core Facility to perform gene expression profiling using the mouse Exon1.0ST Affymetrix platform. Differential expression analysis was carried out on normalized data as described previously (Donati *et al*, 2017).

Gene Set Enrichment Analysis was performed to compare expression signature genes in SG vs IFE and HF (ranked *Z*-score values of differentially expressed genes (DEG); Donati *et al*, 2017) with respect to a gene subset composed of the nearest genes to the ΔNLef1 peaks from Chip-Seq data.

Human Gata6 ChIP-Seq datasets were interrogated using the "Target Genes" module of ChIP-Atlas (Oki *et al*, 2018 and http://chip-atlas.org). This module predicts genes directly regulated by a given protein, based on binding profiles of all public ChIP-Seq data for particular gene loci.

KEGG pathway analysis of DEG in WT vs epidermal specific Gata6 knockout keratinocytes (Wang *et al*, 2017) and Gene Ontology analysis of DEG in WT vs K14ΔNLef1 keratinocytes were performed using DAVID software (https://david.ncifcrf.gov). Heatmaps were generated via Gene-E (http://www.broadinstitute.org). Network visualization was achieved using Cytoscape (http://www.cytoscape.org). Protein schematization was performed using IBS (http://ibs.biocuckoo.org). SNAP2 (Hecht *et al*, 2015) and PredictSNP (Bendl *et al*, 2014) algorithms were used to predict the effect of Gata6 mutations.

### Cell culture and siRNA transfection

Primary mouse keratinocytes were isolated from dorsal skin as previously described (Jensen *et al*, 2010) and cultured on confluent irradiated 3T3-J2 fibroblast feeders in calcium-free FAD medium (DMEM: Ham's F12, 3:1, $1.8 \times 10^{-4}$ M adenine) supplemented with 10% fetal calf serum, hydrocortisone (0.5 μg ml$^{-1}$), insulin (5 μg ml$^{-1}$), cholera toxin (8.4 ng ml$^{-1}$), and epidermal growth

factor (10 ng ml$^{-1}$). siRNAs for the negative control and mouse Lef1 (Ambion) were introduced into cells by nucleofection using the Amaxa 96-well shuttle system (Lonza) as previously described (Mulder *et al*, 2012).

The SebE6E7 sebocyte line was obtained and cultured as described previously (Lo Celso *et al*, 2008). All cell stocks were routinely tested for mycoplasma contamination and were negative. SebE6E7 cells were transfected with a control plasmid or human full-length or ΔN34Lef1 plasmids (Takeda *et al*, 2006) using jetPRIME transfection reagent (Polyplus transfection) following the manufacturer's instructions.

### RT–qPCR

For RT–qPCR, total RNA was isolated using RNeasy kits (Qiagen). cDNAs were generated using the SuperScript III Supermix (Invitrogen) or QuantiTect Reverse Transcription Kit (Qiagen) and analyzed using Power SYBR Green (Applied Biosystems) or SYBR Green Master Mix (Life Technologies) and custom-made primers.

Primer sequences for mouse genes were as follows: Gata6 (forward GGATTCTTGGTGTGCTCTGG and reverse ATTTTTGCTGCCATCTGGAC); Klf4 (forward CTGTCACACTTCTGGCACTGA and reverse GTTCTCATCTCAAGGCACACC); Cd44 (forward CATGGAATACACCTGCGTAGC and reverse CTAGATCCCTCCGTTTCATCC); Lef1 (forward GCTCCTGAAATCCCCACCTTC and reverse GGATGAGGGATGCCAGTTGTG); JunB (forward CCCTGACCCGAAAAGTAGCTG and reverse CTGGCAGCCTGTCTCTACACG); Met (forward CAACCATGAGCACTGTTTCAA and reverse TTTGATGAAGGTGGAGATGGA); Egfl6 (forward TCGATGAATGTGCGTCTAGC and reverse TATTGAGGCAATTGGCATGG); Tiam1 (forward GAGGGCTGTGAGAGGAAATCT and reverse CTGTGGATGAAGATGGCATTC); Wnt16 (forward ACTACCACTTCCACCCAGCTC and reverse CAGGAACATTCGGTCATGTTG); Pparg (forward ATAAAGTCCTTCCCGCTGACC and reverse CTGGCACCCTTGAAAAATTCG); and Tnc (forward TTGCTGGGTCTCAGTTTCATC and reverse ACAGACTCAGCCATCACCAAC). β-actin was used as a housekeeping gene (forward TGGCGTGAGGGAGAGCATAG and reverse GCCAACCGTGAAAAGATGACC).

Primer sequences for human genes were as follows: Gata6 (forward GTGCCCAGACCACTTGCTAT and reverse CCCTGAGGCTGTAGGTTGTG). Gapdh, Rpl13, and Tbp were used as housekeeping genes: Gapdh (forward GAAGAGAGAGACCCTCACTGCTG and reverse ACTGTGAGGAGGGGAGATTCAGT); Rpl13 (forward AACAGCTCATGAGGCTACGG and reverse AACAATGGAGGAAGGGCAGG); and Tbp (forward GTGACCCAGCATCACTGTTTC and reverse GAGCATCTCCAGCACACTCT).

### Microsatellite instability analysis

Microsatellite instability in sebaceous tumors was determined as previously published (Woerner *et al*, 2015; Germano *et al*, 2017). DNA was extracted from paraffin-embedded tumors using a ReliaPrep FFPE gDNA Miniprep System Kit (Promega). Amplification was performed on 20 ng DNA with a Type-it Microsatellite PCR Kit (Qiagen). The cycling profile was as follows: 1 cycle at 95°C for 5 min, 95°C for 30 s, 56°C for 90 s, and 72°C for 30 s for a total of 28 cycles; final extension at 60°C for 30 min. The following labeled primers were used: mBat64, forward fluorescein-GCCCACACTCCTGAAA

ACAGTCAT and reverse CCCTGGTGTGGCAACTTTAAGC; AC096777, forward JOE-TCCCTGTATAACCCTGGCTGACT and reverse GCAACCAGTTGTCCTGGCGTGGA; AA003063, forward Tamra-ACGTCAAAAATCAATGTTAGG and reverse CAGCAAGGGTCCCTGTCTTA; U12235, forward JOE-GCTCATCTTCGTTCCCTGTC and reverse CATTCGGTGGAAAGCTCTGA; and L24372, forward fluorescein-GGGAAGACTGCTTAGGGAAGA and reverse ATTTGGCTTTCAAGCATCCATA. PCR fragments were separated using Applied Biosystems Big-Dye Ver 3.1 chemistry on an Applied Biosystems model 3730 automated capillary DNA sequencer. Raw data were visualized with Geneious software and further analyzed as previously described (Keysselt *et al*, 2017).

### Statistics and reproducibility

Statistical analyses were performed using GraphPad Prism software. No statistical method was used to predetermine sample size. Panels showing images are representative of at least two independent experiments as indicated in each panel of the figure legends.

# Data availability

All data that support the conclusions are available from the authors on request. Microarray and ChIP-Seq data that support the findings of this study have been deposited in the Gene Expression Omnibus (GEO) under accession codes GSE62608 (https://www.ncbi.nlm.nih.gov/geo/query/acc.cgi?acc = GSE62608), GSE118073 (https://www.ncbi.nlm.nih.gov/geo/query/acc.cgi?acc = GSE118073), and GSE118074 (https://www.ncbi.nlm.nih.gov/geo/query/acc.cgi?acc = GSE118074).

Expanded View for this article is available online.

### Acknowledgements

We are grateful to all members of the Watt laboratory for helpful discussions. We thank Beate M Lichtenberger for providing mouse skin samples and Melis Kayikci for support on computational analysis. We thank Raymond J. Cho for providing the data from North *et al* (2018). We thank Rudolf Grosschedl and Ingrid Falk for providing mutant mouse skin. We thank DNA Sequencing & Services (MRC I PPU, School of Life Sciences, University of Dundee, Scotland, www.dnaseq.co.uk) for DNA fragment analysis. We are also grateful for funding from the Department of Health via the National Institute for Health Research comprehensive Biomedical Research Centre award to Guy's & St Thomas' National Health Service Foundation Trust in partnership with King's College London and King's College Hospital NHS Foundation Trust. This work was funded by grants to FMW from the UK Medical Research Council (G1100073) and the Wellcome Trust (096540/Z/11/Z). ER is the recipient of an EMBO Advanced Fellowship (aALTF 523-2017).

### Author contributions

GD and FMW conceived the study and designed the experiments with the help of BO and ER. GD, BO, ER, EH, GG, RF, and RM performed experiments. KN and SQ provided the human samples. GD, BO, ER, EH, GG, and FMW analyzed the data. BO, GD, and FMW wrote the manuscript.

### Conflict of interest

The authors declare that they have no conflict of interest. FMW is currently on secondment as Executive Chair of the UK Medical Research Council.

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
