## [Review Process File · The EMBO Journal]

Mutant Lef1 controls Gata6 in sebaceous gland development and cancer

Bénédicte Oulès, Emanuel Rognoni, Esther Hoste, Georgina Goss, Ryan Fiehler, Ken Natsuga, Sven Quist, Remco Mentink, Giacomo Donati and Fiona M. Watt

Review timeline:	Submission date:	21st Aug 2018
	Editorial Decision:	4th Oct 2018
	Revision received:	14th Jan 2019
	Editorial Decision:	8th Feb 2019
	Revision received:	17th Feb 2019
	Accepted:	19th Feb 2019

Editor: Daniel Klimmeck

Transaction Report:

1st Editorial Decision

4th Oct 2018

Thank you for the submission of your manuscript (EMBOJ-2018-100526) to The EMBO Journal. Thank you also for your patience with my response, which got delayed due to protracted input from one of the referees at this busy time of the year. Your manuscript has been sent to three referees, and we have received reports from all of them, which I enclose below.

As you will see, the referees acknowledge the potential interest and novelty of your work, although they also express a number of major issues that will have to be addressed before they can support publication of your manuscript in The EMBO Journal. In more detail, referee #1 states that your claims on a developmental role of Lef1 as endogenous driver of GATA6 and sebaceous gland differentiation are not sufficiently supported by the current data, which in his/her view undermines the impact of your findings (ref#1, pt.1). In line, this referee asks you to characterize factors restricting temporal and spatial co-occurrence of GATA6 in a subset of Lef1-pos cells (ref#1, pt.2). Referee #2 agrees in that generality and physiological relevance of the Lef1-GATA6 axis has to be more rigorously proven and suggests orthogonal experimental models (ref#2, pt.1). In addition, this reviewer asks you to better address the spatial restrictions and lineage dependencies of GATA6 within the SG (ref#2, pt.2, see also ref#3, pt.1). Finally, the referees point to issues related to experimental design, data schematic representation, statistics, as well as additional controls required that would need to be conclusively addressed to achieve the level of robustness needed for The EMBO Journal.

I judge the comments of the referees to be generally reasonable and given their overall interest, we are in principle happy to invite you to revise your manuscript experimentally to address the referees' comments.

REFeree REPORTS:

Referee #1:

The manuscript by Oulès et al. addresses role of the transcription factor Lef1 in sebaceous gland (SG) development and tumorigenesis. Lef1 is best known for its role as a mediator of the canonical Wnt signaling positively regulating transcription in complex with beta-catenin. Mice expressing an N-terminally truncated Lef1 (i.e. unable to bind beta-catenin, deltaNLef1) had been shown to convert hair follicles (HF) into epidermal cysts/ectopic sebocytes and to spontaneously develop sebaceous adenomas and sebaceomas. Previous studies had revealed the occurrence of Lef1 mutations in SG tumors in humans but how Lef1 contributes to carcinogenesis had remained elusive. Further, the authors had recently shown that that transcription factor Gata6 plays a role in sebaceous lineage determination (its loss leading to 50% reduction in JZ and sebaceous duct cells) and is highly upregulated in junctional zone (the intersection between the HF, interfollicular epidermis, and SG) in mice overexpressing deltaNLef1.

Here, the authors provide evidence linking Lef1 to transcription factor Gata6 in a beta-catenin independent manner. Lineage tracing experiments indicate that the Gata6 lineage gives rise to the upper part of the sebaceous gland, including ducts. In utero overexpression of Gata6 was used to show that Gata6 promotes formation of ectopic SG-like structures. Then the authors move on to show that deletion of Gata6 increases the number of tumors in K14-deltaNLef1 model while decreasing the proportion of sebaceous tumors. The increased tumor burden was linked with the role of Gata6 in DNA mismatch repair via MLH1 and MSH2 genes. Finally, Gata6 was shown to be upregulated in human skin tumors where it was associated with SG-like tumors such that low levels of Gata6 were less differentiated and more aggressive.

The identified beta-catenin independent link between Lef1 and Gata6 is interesting and although the previous study by the same group had already shown upregulation of Gata6 in the deltaNLef1 mouse model (Donati et al., 2017), no further details on the crosstalk were known. In principle, this paper is a nice advancement in the field.

I have a few specific comments on the manuscript:

Major:

1.

It is proposed that Lef1 can drive Gata6 expression in a beta-catenin independent manner. While the independence of beta-catenin is clearer, Lef1 as driver of endogenous Gata6 expression is less clear. Data showing this is qualitative rather than quantitative. Intriguingly, during embryonic development Lef1 and Gata6 overlap only for a short developmental time-window and only in a small subset of hair germ cells. To substantiate their conclusions, the overlap in expression should be quantified (i.e. are % of Gata6+ being Lef1+ and vice versa). Also quantification on the spatial expression would be informative. Are the double positive cells always localized to the "edges" of the Lef1-positive domain? Do the authors have access to Lef1 null embryos? Analysis of Gata6 expression at E15.5 would be informative.

2.

As a follow-up to previous: can the authors give any insights what restricts the expression of Gata6 into a subset of Lef1 cells (embryonically, or in K14-DeltaNLef1 model)? What about Sox9? It is mentioned that there is partial co-expression of Gata6 with Sox9 at P1, but I could not see this in Figure S2. What about E15.5? Showing separate channels and quantifications would be helpful.

3.

It is shown that there is no co-expression of Gata6 with nuclear Tcf3/4, but these data are from P1 mice. What about at E15.5?

4.

Fig.S1 reports Affymetrix data. I could not find information on how many biological replicates were used, nor indication whether the data have been deposited to anywhere. Further, I could not access the other two reported GEO datasets (GSE118073 and GSE118074) for reviewing purposes.

5.

The rationale behind the WT:GFP experiment (Fig. S3A) is not clear.

6.

Lineage tracing of E16.5 Gata6 expressing cells is reported at P13 in tail skin. Please, show also what cells are traced, i.e. what is labeled at E18.5, 2 days after activation of CreERT2. At what developmental stage are tail hair follicles at E16.5?

7.

It is reported that more tumors form in K14-deltaN Δ Lef1 upon Gata6 deletion and that there is a decrease in tumors with SG elements. The authors interpret this to indicate that SG differentiation is significantly reduced. However, this is only in relative terms, and in fact it looks like in absolute numbers (which also should be shown) there is as many, or even more tumors with "SG-elements". An alternative interpretation could be that loss of Gata6 has no impact on tumors with SG-elements but increases the number of papilloma-like tumors. How can the authors tell apart the tumor-suppressor role and the effect of Gata6 in the tumor type? The following statement in Discussion (p. 19) seems like a strong overstatement to me: "Our data clearly indicate that Gata6 expression is responsible for sebaceous differentiation observed in the tumors (Fig 4C)"

8.

The discussion is lengthy. Discussion on the role of Hedgehog pathway (p. 19) seems out of context here and cuts the flow of the text in Discussion.

9.

Some figures lack statistical analysis: 1F, qPCR data in S1. Could the authors explain what the qPCR values are in S1? The gene expressions seem to be normalized to something, but to what?

Minor:

1.

Color labels in Fig 1C would be useful.

Referee #2:

This study by Oules, Watt et al., describes a β -catenin-independent function for Lef1 in activating expression of Gata6. Using ChIP-Seq and RNA-Seq data, the authors find that a dominant negative form of Lef1 (deltaN-Lef1) induces a sebaceous gland gene expression program that includes Gata6. This study further showed that lentiviral Gata6 overexpression was sufficient to induce ectopic sebaceous glands, and that Gata6+ cells comprise a distinct population in the upper sebaceous gland/sebaceous duct region. Importantly, in the absence of Gata6, deltaN-Lef1-driven tumors were more abundant, displayed reduced sebaceous differentiation, and were genomically unstable. Finally, the authors extend this work to a variety of human skin tumors, particularly those with sebaceous differentiation, and report that sebaceous carcinomas with aberrant Gata6 have increased mutation burden and lower MMR genes. Overall, this is an expansive, wide-ranging study that builds upon and extends previous work from the same lab. Although some experiments are redundant from the previous publication, the cancer studies, both in mouse and human, are especially important for understanding how tumors lose differentiation features to become more aggressive. The identification of Gata6 as a marker for tumors that differ in mutation burden/MMR is also a critical advance that will be useful for the field.

Major Comments

1. The claim that the truncated deltaN-Lef1 allele represents a β -catenin-independent mechanism for activating Gata6 expression may be valid, but this may also be a peculiarity specific to this allele. Indeed, although Gata6 is a direct target of deltaN-Lef1 by ChIP-Seq, only a minority of cells that express the transgene upregulate Gata6 in vivo. The authors claim that these effects may be niche-specific and require other factors, which could very well be the case, but it would be useful to confirm this result using an independent approach that does not rely solely on deltaN-Lef1 or activating β -catenin (which leads to a negative, albeit supportive, in vivo result). Possible approaches might include overexpressing full-length Lef1 or a human disease-relevant mutant form of Lef1 found in sebaceous tumors (e.g. from Takeda, Watt et al. in Nat Medicine); or ablating β -catenin - and seeing whether this upregulates Gata6 expression. Either in vitro or in vivo data would

be useful.

2. In their mouse development studies, the authors find that *Lef1* and *Gata6* are co-expressed only during hair follicle development, in a minority of cells that are just above the proximal growing end of the hair germ (Fig. 2B). In their schematics in Fig. 2C,2F, they depict a continuous streak (yellow/red) of differentiated cells extending upward from the hair follicle. In contrast, previous studies by Mesler, Wong et al. in *Cell Reports* indicated that *Gata6* is only expressed later in the upper hair germ, does not overlay with *Lef1*, and that the domain of *Gata6* expression does not extend continuously down to the bottom of the growing hair bud at any time during development.

This distinction is critical, because it helps form the argument that endogenous *Lef1* might activate *Gata6* expression in a cell type that is relevant to the sebaceous lineage. To support their schematic here, the authors need to show a well-sectioned follicle with *Gata6* extending up as a continuous streak of differentiating cells from the base of the hair bud, as they have diagrammed. Continuous staining linking the 2 regions of *Gata6* positivity would suggest a lineage relationship. At this point, though, the images appear to show distinct upper *Gata6*⁺/*Lef1*⁻ and lower *Gata6*⁺/*Lef1*⁺ populations. In the absence of such continuous staining, the authors should revise their schematic, and explain what is the possible significance/fate of the basal layer *Lef1*⁺/*Gata6*⁺ cells, since these cells appear to be positioned lower than the *Lef1*⁻/*Gata6*⁺ cells that will eventually give rise to sebaceous glands or ducts.

Minor Comments

Although mentioned in the Discussion, the authors should point out in the Introduction that Δ N-*Lef1* is thought to act as a dominant negative.

In their GSEA analysis in Fig. 1D/1E, the authors found that Δ N-*Lef1* bound mainly repressed genes from the sebaceous signature. However, they also find that Δ N-*Lef1* can activate SG genes such as *Gata6* in the absence of β -catenin. This is somewhat confusing to the reader, and the authors should clarify (if I'm interpreting this correctly) that Δ N-*Lef1* can possess both the ability to activate genes (*Gata6*) as well as suppress genes (please include a few specific examples), both of which are associated with sebaceous lineage differentiation.

In Fig. 2D, the authors state that *Gata6* is expressed in developing IRS that extends out into the IFE. In fact, Mesler et al., observed that these cells display early companion layer features, or can simply be referred to as early differentiating cells in the developing follicle. This should be clarified, and this study should probably be cited here.

The legend for Fig. 3B should clarify what is depicted in the multiple images. Presumably, the yellow dotted lines indicate a cyst, and this should be defined.

Fig. 5A should say "indirect" instead of "undirect."

The microsatellite analysis in Fig. 5C was confusing, and it's not clear from the figure what is changed.

For Fig. 6B and several other figures showing tumor tissue arrays, the staining for *Gata6* is difficult to see. I would prefer to see fewer, but larger, examples, showing clear representative positive or negative staining, with the rest of the panel moved to supplemental data.

For Fig. 6B, can the authors define in the text what % of cells in the tumor need to be positive for *Gata6*, in order for the entire tumor to be designated as *Gata6*⁺? What is the threshold?

In the Discussion, the authors state that "a reduction in *Gata6* expression is associated with less differentiated and more aggressive UV-related SebC, independent of GATA6 mutational status (Fig. 7 C)." It's not clear to me if data showing *Gata6* mutation status vs. expression were depicted here.

Referee #3:

EMBOJ-2018-100526-T, corr. author Prof Watt

"A Lef1-Gata6 axis governs sebaceous gland development and cancer"

General summary and opinion about the principle significance of the study, its questions and findings

In this manuscript, the authors identify by chIP-seq direct target genes of the N-terminally-truncated form of Lef1 (Δ NLef1) in a mouse model of sebaceous skin tumors. Using two different in vivo mouse models, they show that Gata-6 is a direct transcriptional target of Δ NLef1 that is induced by Lef1 independently of β -catenin. They further showed that Gata-6 promotes sebaceous gland identity in a cell compartment-dependent manner using overexpression of Gata-6 in mouse and lineage tracing experiments. Finally, this study identifies Gata-6 as a tumor-suppressor gene in mouse and human and defines the mechanism by which loss of Gata-6 affect tumorigenesis. Overall, this is a well-controlled and complete study with impressive amount of in vivo data that addresses important questions about what control sebaceous differentiation in development and cancer.

Specific major concerns essential to be addressed to support the conclusions

1. Figure 3: do the ectopic sebaceous glands formed in the G6OE are only composed of upper sebaceous gland and sebaceous ducts? If no, then these data do not support the conclusion that Gata 6 lineage is responsible for generating only the upper part of the sebaceous glands, including the sebaceous ducts.

Minor concerns that should be addressed

1. In figure 2G: it is unclear what are the placode cells as the figure is showing a PEG stage. Please clarify
2. The Figure 2I is not convincing as only epidermis is shown compared to the WT control. Please provide the same size image of the WT and K14Cre/bcat Flox(ex3)/+ mice.
3. Figure S3: the use of the WT: GFP chimeric mice should be removed from this study as the conclusions can only be speculative and cannot prove that sebaceous glands progenitors exclusively populate the upper sebaceous gland and the ducts.
4. A Summary model for the role of GATA6 as a tumor suppressor gene should be included
5. The schematic representation of GATA6+ cells in human sebaceous gland in Figure 6 is confusing and is not helpful to clarify the message.
6. For non-skin experts please label sweat glands into the figure as 'top right panel' mentioned in the legends is not enough to find the structures.

1st Revision - authors' response

14th Jan 2019

Referee #1:

The manuscript by Oulès et al. addresses role of the transcription factor Lef1 in sebaceous gland (SG) development and tumorigenesis. Lef1 is best known for its role as a mediator of the canonical Wnt signaling positively regulating transcription in complex with beta-catenin. Mice expressing an N-terminally truncated Lef1 (i.e. unable to bind beta-catenin, Δ NLef1) had been shown to convert hair follicles (HF) into epidermal cysts/ectopic sebocytes and to spontaneously develop sebaceous adenomas and sebaceomas. Previous studies had revealed the occurrence of Lef1 mutations in SG tumors in humans but how Lef1 contributes to carcinogenesis had remained elusive. Further, the authors had recently shown that that transcription factor Gata6 plays a role in sebaceous lineage determination (its loss leading to 50% reduction in JZ and sebaceous duct cells)

and is highly upregulated in junctional zone (the intersection between the HF, interfollicular epidermis, and SG) in mice overexpressing deltaNLef1.

Here, the authors provide evidence linking Lef1 to transcription factor Gata6 in a beta-catenin independent manner. Lineage tracing experiments indicate that the Gata6 lineage gives rise to the upper part of the sebaceous gland, including ducts. In utero overexpression of Gata6 was used to show that Gata6 promotes formation of ectopic SG-like structures. Then the authors move on to show that deletion of Gata6 increases the number of tumors in K14-deltaNLef1 model while decreasing the proportion of sebaceous tumors. The increased tumor burden was linked with the role of Gata6 in DNA mismatch repair via MLH1 and MSH2 genes. Finally, Gata6 was shown to be upregulated in human skin tumors where it was associated with SG-like tumors such that low levels of Gata6 were less differentiated and more aggressive.

The identified beta-catenin independent link between Lef1 and Gata6 is interesting and although the previous study by the same group had already shown upregulation of Gata6 in the deltaNLef1 mouse model (Donati et al., 2017), no further details on the crosstalk were known. In principle, this paper is a nice advancement in the field.

I have a few specific comments on the manuscript:

Major:

1. It is proposed that Lef1 can drive Gata6 expression in a beta-catenin independent manner. While the independence of beta-catenin is clearer, Lef1 as driver of endogenous Gata6 expression is less clear. Data showing this is qualitative rather than quantitative. Intriguingly, during embryonic development Lef1 and Gata6 overlap only for a short developmental time-window and only in a small subset of hair germ cells. To substantiate their conclusions, the overlap in expression should be quantified (i.e. are % of Gata6+ being Lef1+ and vice versa). Also quantification on the spatial expression would be informative. Are the double positive cells always localized to the "edges" of the Lef1-positive domain? Do the authors have access to Lef1 null embryos? Analysis of Gata6 expression at E15.5 would be informative.

To answer this point, we performed new immunostaining for Gata6 and Lef1 in back skin from wild-type embryos ranging from E15.5 to E18.5 and used the hair follicle (HF) atlas published in Paus et al JID 1999 to identify the different HF stages (new Fig 2A). We were unable to detect reproducible expression of Gata6 before HF stage 4. At this stage, Gata6 was expressed in cells in the upper part of the HF where the sebaceous glands (SG) will form at stage 5. We were unable to identify cells co-expressing Lef1 and Gata6 in embryonic or adult skin. There was no difference in Gata6 expression in developing guard (starting around E14), anagen (starting at E16-E17) or zig-zag (starting at E18) hairs. We also examined the skin of E18.5 Lef1 null embryos and saw no abnormal Gata6 expression (new Fig 2B).

We think that the discrepancy between our new results and our previous findings comes from differences in sample processing. The previous staining was performed on paraffin-embedded sections and we believe that autofluorescence and fixation artefacts led to misinterpretation of the images. We are deeply grateful to our reviewers for questioning the staining and thereby saving us from considerable embarrassment. We have corrected the text and added representative images of Gata6 and Lef1 expression at different HF stages in new Fig 2A. We have also added new data showing that Gata6 expression during physiological HF morphogenesis is similar in human and mouse skin (Fig EV2C).

2. As a follow-up to previous: can the authors give any insights what restricts the expression of Gata6 into a subset of Lef1 cells (embryonically, or in K14-DeltaNLef1 model)? What about Sox9? It is mentioned that there is partial co-expression of Gata6 with Sox9 at P1, but I could not see this in Figure S2. What about E15.5? Showing separate channels and quantifications would be helpful.

We now provide additional co-staining of Gata6 and Sox9 in back skin from wild-type embryos (new Fig. 2A). As previously described, a subset of Sox9+ keratinocytes will give rise to sebocytes (Nowak JA et al Cell Stem Cell 2008, Frances D & Niemann C Dev Biol 2012). During HF morphogenesis, we now show that Gata6 is expressed in some Sox9+ cells and marks a population

of early progenitors of the sebaceous glands. As HF develop further, the proportion of double positive Sox9/Gata6 cells becomes negligible, coinciding with the restriction of Sox9 expression to bulge cells (Nowak JA et al Cell Stem Cell 2008). It is noteworthy that Sox9 expression and Wnt/b-catenin signalling are mutually exclusive in developing HF (Xu Z et al eLife 2015). Therefore, we speculate that during HF morphogenesis Gata6 is expressed through Sox9 induction, independent of Wnt/b-catenin signalling.

3. It is shown that there is no co-expression of Gata6 with nuclear Tcf3/4, but these data are from P1 mice. What about at E15.5?

We have not included the data because the relevance is now in doubt, given the lack of co-expression of Gata6 and Lef1.

4. Fig.S1 reports Affymetrix data. I could not find information on how many biological replicates were used, nor indication whether the data have been deposited to anywhere. Further, I could not access the other two reported GEO datasets (GSE118073 and GSE118074) for reviewing purposes.

For Fig. S1 Affymetrix data (new FigEV1), 3 biological replicates were analysed. These data are deposited in GSE118073, while the GSE118074 dataset corresponds to the genome-wide maps of Δ NLef1 in primary mouse keratinocytes. We apologize for failing to provide the tokens to access our GEO datasets. The tokens are wvizouearjolfqd for GSE118073 and qvczkwesrbihvix for GSE118074.

5. The rationale behind the WT:GFP experiment (Fig. S3A) is not clear.

We now explain more clearly that the chimera experiment provides evidence for an upper SG progenitor. We then go on to identify it as being the Gata6+ lineage.

6. Lineage tracing of E16.5 Gata6 expressing cells is reported at P13 in tail skin. Please, show also what cells are traced, i.e. what is labeled at E18.5, 2 days after activation of CreERT2. At what developmental stage are tail hair follicles at E16.5?

As mentioned above, for clarity we now present our data as HF stages rather than embryonic ages. We have repeated the lineage tracing experiment inducing recombination in pregnant females at E16.5 and E18.5. In line with our new Gata6 embryonic expression analysis (new Fig 2A) the first cells to be labelled are located in the upper part of stage 4 HF. The Gata6Cre cell labelling is strictly HF stage-dependent and thus HF labelling efficiency correlates with the abundance of stage 4 HF at the different recombination time points. We have now included this information in the text and show representative pictures of labelled HF at stage 4 and 5, two days after recombination (new Fig EV3B).

7. It is reported that more tumors form in K14- Δ NLef1 upon Gata6 deletion and that there is a decrease in tumors with SG elements. The authors interpret this to indicate that SG differentiation is significantly reduced. However, this is only in relative terms, and in fact it looks like in absolute numbers (which also should be shown) there is as many, or even more tumors with "SG-elements". An alternative interpretation could be that loss of Gata6 has no impact on tumors with SG-elements but increases the number of papilloma-like tumors. How can the authors tell apart the tumor-suppressor role and the effect of Gata6 in the tumor type? The following statement in Discussion (p. 19) seems like a strong overstatement to me: "Our data clearly indicate that Gata6 expression is responsible for sebaceous differentiation observed in the tumors (Fig 4C)"

As described in Fig 4C legend, a total of 268 tumours were analysed. From these 268 tumours, 143 were found in 7 K14 Δ NLef1 mice (total of 103 sebaceous tumours and 40 papillomas), while 125 were found in 5 K14 Δ NLef1:cKO mice (total of 55 sebaceous tumours and 70 papillomas). We now present the average number of each tumour type per mouse in Fig 4C. This shows that not only does Gata6 deletion increase the tumour burden in K14 Δ NLef1 mice, it also reduces the number of tumours with sebaceous differentiation. We changed the sentence in the Discussion to avoid overstatement.

8. The discussion is lengthy. Discussion on the role of Hedgehog pathway (p. 19) seems out of context here and cuts the flow of the text in Discussion.

We have removed this section from the Discussion.

9. Some figures lack statistical analysis: 1F, qPCR data in S1. Could the authors explain what the qPCR values are in S1? The gene expressions seem to be normalized to something, but to what?

We have added statistical analysis to Fig 1F and Fig. S1 (now new Fig EV1B). In Fig EV1B, the qPCR data were generated from sorted cells as in Fig EV1A and all mouse qPCR values were normalised to b-actin as a housekeeping gene.

Minor:

1. Color labels in Fig 1C would be useful.

We have added colour labels to Fig 1C.

Referee #2:

This study by Oules, Watt et al., describes a β -catenin-independent function for Lef1 in activating expression of Gata6. Using ChIP-Seq and RNA-Seq data, the authors find that a dominant negative form of Lef1 (deltaN-Lef1) induces a sebaceous gland gene expression program that includes Gata6. This study further showed that lentiviral Gata6 overexpression was sufficient to induce ectopic sebaceous glands, and that Gata6⁺ cells comprise a distinct population in the upper sebaceous gland/sebaceous duct region. Importantly, in the absence of Gata6, deltaN-Lef1-driven tumors were more abundant, displayed reduced sebaceous differentiation, and were genomically unstable. Finally, the authors extend this work to a variety of human skin tumors, particularly those with sebaceous differentiation, and report that sebaceous carcinomas with aberrant Gata6 have increased mutation burden and lower MMR genes. Overall, this is an expansive, wide-ranging study that builds upon and extends previous work from the same lab. Although some experiments are redundant from the previous publication, the cancer studies, both in mouse and human, are especially important for understanding how tumors lose differentiation features to become more aggressive. The identification of Gata6 as a marker for tumors that differ in mutation burden/MMR is also a critical advance that will be useful for the field.

Major Comments

1. The claim that the truncated deltaN-Lef1 allele represents a β -catenin-independent mechanism for activating Gata6 expression may be valid, but this may also be a peculiarity specific to this allele. Indeed, although Gata6 is a direct target of deltaN-Lef1 by ChIP-Seq, only a minority of cells that express the transgene upregulate Gata6 in vivo. The authors claim that these effects may be niche-specific and require other factors, which could very well be the case, but it would be useful to confirm this result using an independent approach that does not rely solely on deltaN-Lef1 or activating β -catenin (which leads to a negative, albeit supportive, in vivo result). Possible approaches might include overexpressing full-length Lef1 or a human disease-relevant mutant form of Lef1 found in sebaceous tumors (e.g. from Takeda, Watt et al. in Nat Medicine); or ablating β -catenin - and seeing whether this upregulates Gata6 expression. Either in vitro or in vivo data would be useful.

To answer this point, we overexpressed human full-length Lef1 or Δ N34Lef1 in the SebE67 human sebocyte cell line (new Fig 1G) as performed in Takeda H et al Nat Med 2006. Δ N34Lef1 expression led to a significant induction of Gata6. Full-length Lef1 expression also led to an induction of Gata6, although the effect was not statistically significant.

2. In their mouse development studies, the authors find that Lef1 and Gata6 are co-expressed only

during hair follicle development, in a minority of cells that are just above the proximal growing end of the hair germ (Fig. 2B). In their schematics in Fig. 2C,2F, they depict a continuous streak (yellow/red) of differentiated cells extending upward from the hair follicle. In contrast, previous studies by Mesler, Wong et al. in Cell Reports indicated that Gata6 is only expressed later in the upper hair germ, does not overlay with Lef1, and that the domain of Gata6 expression does not extend continuously down to the bottom of the growing hair bud at any time during development.

This distinction is critical, because it helps form the argument that endogenous Lef1 might activate Gata6 expression in a cell type that is relevant to the sebaceous lineage. To support their schematic here, the authors need to show a well-sectioned follicle with Gata6 extending up as a continuous streak of differentiating cells from the base of the hair bud, as they have diagrammed. Continuous staining linking the 2 regions of Gata6 positivity would suggest a lineage relationship. At this point, though, the images appear to show distinct upper Gata6+/Lef1(-) and lower Gata6+/Lef1+ populations. In the absence of such continuous staining, the authors should revise their schematic, and explain what is the possible significance/fate of the basal layer Lef1+/Gata6+ cells, since these cells appear to be positioned lower than the Lef1(-)/Gata6+ cells that will eventually give rise to sebaceous glands or ducts.

As discussed in response to Referee 1, we no longer observe coexpression of Lef1 and Gata6 during HF morphogenesis. We have therefore removed the schematic. Our new data fully agree with Mesler AL et al Cell Reports 2017.

Minor Comments

Although mentioned in the Discussion, the authors should point out in the Introduction that deltaN-Lef1 is thought to act as a dominant negative.

We have added a sentence about this to the Introduction.

In their GSEA analysis in Fig. 1D/1E, the authors found that deltaN-Lef1 bound mainly repressed genes from the sebaceous signature. However, they also find that deltaN-Lef1 can activate SG genes such as Gata6 in the absence of β -catenin. This is somewhat confusing to the reader, and the authors should clarify (if I'm interpreting this correctly) that deltaN-Lef1 can possess both the ability to activate genes (Gata6) as well as suppress genes (please include a few specific examples), both of which are associated with sebaceous lineage differentiation.

We have clarified in the text that Δ NLef1 can indeed activate or repress genes associated with sebaceous differentiation. To support this, we performed a new analysis by intersecting Δ NLef1 direct targets genes with DEG in SG vs IFE and HF, and with K14 Δ NLef1 vs WT epidermis expression profile (Fig. EV1D). In addition to Gata6 (Fig 1C), we show that Δ NLef1 directly upregulates Pparg and Edar, two receptors that are well characterised as SG positive regulators (Niemann C and Horsley V Semin Cell Dev Biol 2012, Chang SH Plos One 2009) and together with Aadac and Igfbp2 expressed in SG (<http://linnarssonlab.org/epidermis/> and Joost S et al Cell Systems 2016). In parallel, Δ NLef1 directly down-regulates genes such as Klf5, important for IFE identity (Ge Y Cell 2017), Igfbp3 that is expressed in SG (Dahlhoff M Exp Dermatol 2016), and Slco2a1. Loss of function mutations in Slco2a1 are associated with multi-organ disease including sebaceous hyperplasia (Guo T Mol Med Rep 2017).

In Fig. 2D, the authors state that Gata6 is expressed in developing IRS that extends out into the IFE. In fact, Mesler et al., observed that these cells display early companion layer features, or can simply be referred to as early differentiating cells in the developing follicle. This should be clarified, and this study should probably be cited here.

We have clarified this point and cited Mesler AL et al Cell Reports 2017.

The legend for Fig. 3B should clarify what is depicted in the multiple images. Presumably, the yellow dotted lines indicate a cyst, and this should be defined.

We apologise for not indicating this in the legend. The yellow dotted lines indeed show a cyst. We have modified the legend accordingly.

Fig. 5A should say "indirect" instead of "undirect."

We have corrected this spelling mistake.

The microsatellite analysis in Fig. 5C was confusing, and it's not clear from the figure what is changed.

We have modified the old Fig 5C to better visualise the graphs. Traces from K14 Δ NLef1 and K14 Δ NLef1:cKO tumours are now superimposed so that it is easier to appreciate the allele shift in some of these microsatellites. It is acknowledged in the literature (Woerner SM et al Mol Carcinogenesis 2015) that microsatellite instability (MSI) analysis in mice usually shows only subtle changes in microsatellite length, in contrast to MSI analysis in humans.

For Fig. 6B and several other figures showing tumor tissue arrays, the staining for Gata6 is difficult to see. I would prefer to see fewer, but larger, examples, showing clear representative positive or negative staining, with the rest of the panel moved to supplemental data.

We have modified the figures containing tumour tissue arrays accordingly.

For Fig. 6B, can the authors define in the text what % of cells in the tumor need to be positive for Gata6, in order for the entire tumor to be designated as Gata6+? What is the threshold?

We now provide the percentage of Gata6-positive cells in each tumour analysed (except for those tumour types in which none of the samples were Gata6-positive) in Fig 6B. We did not use any threshold percentage. Therefore, a tumour was called Gata6-positive if there were any positive cells. The least positive sample contained approximately 3% Gata6-positive cells.

In the Discussion, the authors state that "a reduction in Gata6 expression is associated with less differentiated and more aggressive UV-related SebC, independent of GATA6 mutational status (Fig. 7 C)." It's not clear to me if data showing Gata6 mutation status vs. expression were depicted here.

Fig 7C represents Gata6 expression levels in the 3 different subtypes of sebaceous carcinomas described in North JP et al Nat Comm 2018. It is not a graph showing Gata6 expression vs mutation status. We have now clarified this in the text.

Referee #3:

EMBOJ-2018-100526-T, corr. author Prof Watt "A Lef1-Gata6 axis governs sebaceous gland development and cancer"

General summary and opinion about the principle significance of the study, its questions and findings

In this manuscript, the authors identify by chIP-seq direct target genes of the N-terminally-truncated form of Lef1 (Δ NLef1) in a mouse model of sebaceous skin tumors. Using two different in vivo mouse models, they show that Gata-6 is a direct transcriptional target of Δ NLef1 that is induced by Lef1 independently of β -catenin. They further showed that Gata-6 promotes sebaceous gland identity in a cell compartment-dependent manner using overexpression of Gata-6 in mouse and lineage tracing experiments. Finally, this study identifies Gata-6 as a tumor-suppressor gene in mouse and human and defines the mechanism by which loss of Gata-6 affect tumorigenesis. Overall, this is a well-controlled and complete study with impressive amount of in vivo data that addresses important questions about what control sebaceous differentiation in development and cancer.

Specific major concerns essential to be addressed to support the conclusions

1. Figure 3: do the ectopic sebaceous glands formed in the G6OE are only composed of upper sebaceous gland and sebaceous ducts? If no, then these data do not support the conclusion that Gata

6 lineage is responsible for generating only the upper part of the sebaceous glands, including the sebaceous ducts.

We previously used the same lentiviruses (as those in Figs 3B and C) to test the effect of Gata6 overexpression in primary murine keratinocytes. We demonstrated that Gata6 induces a sebaceous duct/gland (SD/SG) transcriptional program upon differentiation *in vitro*. In addition to SD genes, Gata6 triggered the expression of sebocyte differentiation genes such as Pparg or Fasn while it downregulated genes associated with the androgen receptor gene signature that is a distinctive feature of the lower SG (Donati G et al Nat Cell Biol 2017).

The aim of the experiment presented in Fig 3B was to address the effect of Gata6 *in utero* overexpression on the upper SG compartment. *In utero* overexpression of Gata6 led to an upregulation of the sebocyte marker Fasn in the absence of lipid production (Figs 3B and C). This is consistent with the absence of Gata6 staining in differentiated LipidTOX-positive sebocytes in mouse skin (Donati et al Nat Cell Biol 2017).

Minor concerns that should be addressed

1. In figure 2G: it is unclear what are the placode cells as the figure is showing a PEG stage. Please clarify

We apologise for the unclear labelling in old Fig. 2G. We now provide representative images of each stage of early HF morphogenesis in new Fig 2A.

2. The Figure 2I is not convincing as only epidermis is shown compared to the WT control. Please provide the same size image of the WT and K14Cre/bcat Flox(ex3)/+ mice.

We now provide the same size images showing epidermis and dermis in the new Fig 2D.

3. Figure S3: the use of the WT: GFP chimeric mice should be removed from this study as the conclusions can only be speculative and cannot prove that sebaceous glands progenitors exclusively populate the upper sebaceous gland and the ducts.

We have retained the data and provide a better justification for inclusion.

4. A Summary model for the role of GATA6 as a tumor suppressor gene should be included

We have made a schematic to summarise our work, including the tumour suppressor role of Gata6 (synopsis).

5. The schematic representation of GATA6+ cells in human sebaceous gland in Figure 6 is confusing and is not helpful to clarify the message.

We have removed this schematic.

6. For non-skin experts please label sweat glands into the figure as 'top right panel' mentioned in the legends is not enough to find the structures.

We have modified the new Fig 6A and its legend accordingly.

2nd Editorial Decision

8th Feb 2019

Thank you for submitting the revised version of your manuscript. Your revised study has now been re-evaluated by the three original referees, please find their comments enclosed below. As you will see the referees find that their concerns have been sufficiently addressed and they are now broadly in favour of publication.

Thus, we are pleased to inform you that your manuscript has been accepted in principle for

publication in The EMBO Journal, pending some minor issues regarding discussion, formatting and data representation, as outlined below, which need to be adjusted at re-submission.

REFeree REPORTS :

Referee #1:

I am satisfied with the revisions the authors have made.

Referee #2:

The authors have addressed my concerns in their revised manuscript. This study uses a variety of techniques to explore the connection (or lack thereof) between Lef1, mutant deltaN-Lef1 and Gata6 in development and cancer. Pending a few minor edits to the text, I feel this manuscript is suitable for publication in EMBOJ.

Minor edits:

- Fig. 1G. The authors overexpressed WT Lef1 or deltaN-Lef1 in sebocytes, and observed increased Gata6. Please clarify whether the deltaN-Lef1 allele used here (referred to as deltaN34 in the Methods) is the same allele as the one used for overexpression in mice, or other experiments.

- Fig. EV3A. Only 5 labeling categories are illustrated, but many more staining examples are shown. Please simplify, either by showing only the best example to represent each category; or labeling the various images to show how they were classified.

- Fig. EV3C. The dotted line is off in the upper left image.

- Discussion: The authors attribute Gata6's tumor suppressive ability mainly to its effects on MMR gene expression, but their data also suggest that loss of Gata6 reduces differentiation in the tumor (less SG ductal differentiation in Fig. 4D). Therefore, loss of Gata6 might conceivably reduce differentiation and increase proliferation in the tumor, indirectly leading to increased microsatellite instability. It would therefore be useful if the authors also mentioned this possibility.

- References are out of order for Lo Celso et al., and Van Genderen et al.

Referee #3:

The authors have now answered all the comments I initially raised and they improved the manuscript. I therefore consider that this article should be published in EMBO journal

2nd Revision - authors' response

17th Feb 2019

All requested editorial changes have been made.

Referee #2:

The authors have addressed my concerns in their revised manuscript. This study uses a variety of techniques to explore the connection (or lack thereof) between Lef1, mutant deltaN-Lef1 and Gata6 in development and cancer. Pending a few minor edits to the text, I feel this manuscript is suitable for publication in EMBOJ.

Minor edits:

- Fig. 1G. The authors overexpressed WT Lef1 or deltaN-Lef1 in sebocytes, and observed increased Gata6. Please clarify whether the deltaN-Lef1 allele used here (referred to as deltaN34 in the Methods) is the same allele as the one used for overexpression in mice, or other experiments.

In Fig 1G we transfected a DN34Lef1 allele into human sebocytes. This is the human ortholog of murine DN32Lef1, which is expressed in K14DNLef1 transgenic mice. We clarified this in the legend of Fig 1G.

- Fig. EV3A. Only 5 labeling categories are illustrated, but many more staining examples are shown. Please simplify, either by showing only the best example to represent each category; or labeling the various images to show how they were classified.

We have now labelled all the different images in Figure EV3A.

- Fig. EV3C. The dotted line is off in the upper left image.

We have now correctly positioned the dotted line.

- Discussion: The authors attribute Gata6's tumor suppressive ability mainly to its effects on MMR gene expression, but their data also suggest that loss of Gata6 reduces differentiation in the tumor (less SG ductal differentiation in Fig. 4D). Therefore, loss of Gata6 might conceivably reduce differentiation and increase proliferation in the tumor, indirectly leading to increased microsatellite instability. It would therefore be useful if the authors also mentioned this possibility.

We thank the reviewer for pointing this out. We also think this may contribute to the tumor suppressor role of Gata6 and have now referred to it in the Discussion.

- References are out of order for Lo Celso et al., and Van Genderen et al.

We have now corrected this.

Corresponding Author Name: Fiona M. Watt

Journal Submitted to: EMBO J

Manuscript Number: EMBOJ-2018-100526